# Cryo-EM structures of the human PA200 and PA200-20S complex reveal regulation of proteasome gate opening and two PA200 apertures

**Hongxin Guan**[1☯], **Youwang Wang**[2,3☯], **Ting Yu**[1☯], **Yini Huang**[1,4☯], **Mianhuan Li**[1,5], **Abdullah F. U. H. Saeed**[1], **Vanja Perčulija**[1], **Daliang Li**[1], **Jia Xiao**[1,5], **Dongmei Wang**[1], **Ping Zhu**[2,3]*, **Songying Ouyang**[1,2,4]*

**1** The Key Laboratory of Innate Immune Biology of Fujian Province, Provincial University Key Laboratory of Cellular Stress Response and Metabolic Regulation, Biomedical Research Center of South China, Key Laboratory of OptoElectronic Science and Technology for Medicine of Ministry of Education, College of Life Sciences, Fujian Normal University, Fuzhou, China, **2** National Laboratory of Biomacromolecules, CAS Center for Excellence in Biomacromolecules, Institute of Biophysics, Chinese Academy of Sciences, Beijing, China, **3** University of Chinese Academy of Sciences, Beijing, China, **4** Laboratory for Marine Biology and Biotechnology, Pilot National Laboratory for Marine Science and Technology (Qingdao), Qingdao, China, **5** Institute of Clinical Sciences, The First Affiliated Hospital of Jinan University, Guangzhou, China

☯ These authors contributed equally to this work.
* ouyangsy@fjnu.edu.cn (SO); zhup@ibp.ac.cn (PZ)

**Data Availability Statement:** All relevant data are within the paper and its Supporting Information files. All cryo-EM structure files are available from the EMDB database (accession numbers EMD-

## Abstract

Proteasomes are highly abundant and conserved protease complexes that eliminate unwanted proteins in the cells. As a single-chain ATP-independent nuclear proteasome activator, proteasome activator 200 (PA200) associates with 20S core particle to form proteasome complex that catalyzes polyubiquitin-independent degradation of acetylated histones, thus playing a pivotal role in DNA repair and spermatogenesis. Here, we present cryo–electron microscopy (cryo-EM) structures of the human PA200-20S complex and PA200 at 2.72 Å and 3.75 Å, respectively. PA200 exhibits a dome-like architecture that caps 20S and uses its C-terminal YYA (Tyr-Tyr-Ala) to induce the α-ring rearrangements and partial opening of the 20S gate. Our structural data also indicate that PA200 has two openings formed by numerous positively charged residues that respectively bind (5,6)-bisdiphosphoinositol tetrakisphosphate (5,6[PP]2-InsP$_4$) and inositol hexakisphosphate (InsP$_6$) and are likely to be the gates that lead unfolded proteins through PA200 and into the 20S. Besides, our structural analysis of PA200 found that the bromodomain (BRD)-like (BRDL) domain of PA200 shows considerable sequence variation in comparison to other human BRDs, as it contains only 82 residues because of a short ZA loop, and cannot be classified into any of the eight typical human BRD families. Taken together, the results obtained from this study provide important insights into human PA200-induced 20S gate opening for substrate degradation and the opportunities to explore the mechanism for its recognition of H4 histone in acetylation-mediated proteasomal degradation.

0780 and EMD-0781). All coordinate and structure factor files are available from the PDB database (accession numbers 6KWX and 6KWY).

**Funding:** SO was funded by National Natural Science Foundation of China grants (31770948, 31570875), the high-level personnel introduction grant of Fujian Normal University (Z0210509); PZ was funded by National Natural Science Foundation of China grants (31425007, 31730023), grants from the Chinese Ministry of Science and Technology (2015CB856200, 2017YFA0504700), Special Funds of the Central Government Guiding Local Science and Technology Development (2017L3009), Strategic Priority Research Program from Chinese Academy of Sciences (XDB08010100); HG was funded by National Natural Science Foundation of China grants (31900879), Natural Science Foundation of Fujian province grants (2019J05064). The funders had no role in study design, data collection and analysis, decision to publish, or preparation of the manuscript.

**Competing interests:** The authors have declared that no competing interests exist.

**Abbreviations:** 5,6[PP]2-InsP$_4$, (5,6)-bisdiphosphoinositol tetrakisphosphate; aa, amino acid; AC, acetylated; Blm10, Bleomycin resistance 10; BRD, bromodomain; BRDL, BRD-like; CP, core particle; cryo-EM, cryo–electron microscopy; DBS, double-strand breaks; DTT, Dithiothreitol; FSC, Fourier shell correlation; HbYX, hydrophobic-tyrosine-other; HDAC, histone deacetylase; HPLC-MS, high-performance liquid chromatography–mass spectrometry; HR, HEAT repeat; Ins(1,4,5,6)P$_4$, D-myo-inositol-(1,4,5,6)-tetrakissphosphate; InsP$_6$, inositol hexakisphosphate; Kac, acetyl-lysine; NS-TEM, negative-stain transmission electron microscopy; PA, proteasome activator; PDB, Protein Data Bank; PfPA28, PA28 from *Plasmodium falciparum*; RMSD, root-mean-square deviation; SMRT, silencing mediator of retinoid acid and thyroid hormone receptors.

## Introduction

Eukaryotic cells maintain an intricate and sophisticated system for the removal of cellular proteins in a timely manner. This regulation is carried out primarily by proteasomes [1–3]. Proteasomes are essential multicatalytic proteases located within cytosol and nucleus that degrade various intracellular proteins [4,5]. In mammalian cells, proteasome activators (PAs) form a variety of complexes via binding of the 20S core particle (CP) with one or two activators or regulators belonging to four different types [4, 6–8]. The most broadly conserved type is the eukaryotic 19S activator (regulatory particle/PA700) [9–11]. This factor is an ATP-dependent activator that binds and unfolds ubiquitin-conjugated proteins to promote proteasomal degradation of its substrates [11]. The other three activator families—i.e., the 11S (PA28α/β), PA28γ (also referred to as REGγ), and PA200—are ATP-independent and less broadly conserved than the ATP-dependent activators [2,12–14]. Typically, proteasomes in mammalian cells come in two predominant forms: the activator-unbound 20S proteasomes (CP) and 26S holo-enzymes (CP-19S complex). These two forms account for 50% and 30% of all cellular proteasomes, respectively [15,16]. The remaining 20S proteasome pool associates with other PAs, such as the cytosolic PA28α/β, the nuclear PA28γ and PA200 activators, which enhance the proteolytic capacity of the complex in an ATP- and ubiquitin-independent manner [15]. The proportion of different forms of proteasomes differs in cellular tissues. The proteasomes originating from testicular tissue represent a particular case, with 90% of the proteasomal complexes containing one or two PA200 molecules [17,18]. These non-ATPase activators do not promote general proteolysis of intact globular proteins; instead, they may play a role in a pathway alternative to the canonical proteasome degradation pathway that would allow more efficient protein degradation under certain situations such as metabolic adaptation and stress response [4,15,19–21].

PA200 is a 200-kDa monomeric nuclear protein involved in spermatogenesis, DNA double-strand breaks (DSBs) repair, maintenance of mitochondrial inheritance, and proteasome maturation and assembly [15,17]. Binding of PA200 protomers to 20S CP generates the asymmetric single-capped PA200-CP complex or the symmetric double-capped PA200$_2$-CP, both of which enhance proteasomal degradation of small peptides [22]. However, the mechanism by which the substrate enters the PA200-20S complex is still unclear. Previous research suggested that the substrate protein is either bound within the PA200/Bleomycin resistance 10 (Blm10) dome prior to the association with 20S or requires an assistance of a yet unidentified ATPase [21]. Although the earlier reported structure of Blm10 (the yeast ortholog of human PA200 with sequence identity of less than 20%)-20S complex has shown an opening (13 × 22 Å) on the Blm10 dome (Protein Data Bank [PDB]: 4V7O), it still needs to be determined whether unfolded proteins are able to access the proteasome through this pore because of the lack of the electron density at the outlining of the opening [21]. To partially or fully open the 20S CP gate for substrate entry, Blm10 and PAN/19S activator utilize the penultimate tyrosine/phenylalanine residue in their C-terminal to displace the Pro17 reverse turns of proteasome subunits. Alternatively, 11S activators use an internal "activation loop" to induce the gate opening [6,7,21].

Although proteasomes generally catalyze ATP- and ubiquitin-dependent proteolysis, the proteasomes containing the activator PA200 can catalyze the ubiquitin-independent degradation of acetylated (AC) histones by using its bromodomain (BRD)-like (BRDL) domain for recognition and binding [18,23]. A study of the human BRD family has shown that there are eight distinct BRD families, signifying 61 different BRD variants from 46 distinct proteins [24]. PA200 and Blm10 share almost no sequence homology with any known BRDs, but the BRDL regions of Blm10 structurally resemble BRD [18]. In general, affinities of acetyl-lysine

(Kac) for BRDs are low, especially in case of a single Kac site [24]. Previous studies demonstrated that the BRDL domain of PA200 and Blm10 could bind naturally AC histones from HeLa cells rather than the bacterially expressed N-terminal histone H4 peptides that include a single acetylation site at K16. Thus, the BRDL of Blm10 and PA200 possibly bind acetyl-histones carrying additional posttranslational modifications of histones in vitro [18].

So far, the function of PA200 is not entirely understood. It is unclear whether PA200 functions solely as a proteasome regulator or might harbor independent roles as well [15,17,23]. Besides, the absence of the high-resolution structures of PA200 and PA200-20S makes it challenging to understand the mechanism of 20S CP gate opening triggered by an interaction between PA200 and 20S CP. Similarly, it is still unclear whether there is an opening in the PA200 and whether the substrate (or product) traffic proceeds through PA200 dome and into the 20S CP via this opening. Currently, it is known that PA200 significantly promotes ATP-independent proteasomal degradation of the AC core histones. Still, it is uncertain how PA200 binds unfolded protein and specifically targets AC histones for degradation.

Here, in an effort to understand the mechanism underlying interaction between PA200 and 20S, we used cryo–electron microscopy (cryo-EM) to characterize the 2.72 Å–resolution structure of PA200-20S CP complex and 3.75 Å–resolution structure of PA200. Our complex structure demonstrates how PA200 C-terminal residues bind the proteasome. Remarkably, this interaction is analogous to that of Blm10, 11S, and PAN/19S activators, suggesting a uniform model for interactions between these PAs and 20S that mediate gate opening. Our structures show that PA200 has two openings distinct from Blm10. One of the openings sits on the center of the dome, whereas the other one is located at the edge of PA200. Surprisingly, both of the openings are significantly obstructed by well-defined densities, which are surrounded by dense clusters of positively charged residue side chains of PA200. To reveal the densities, we tried repetitively to determine the small molecules and eventually confirmed that the two small molecules are (5,6)-bisdiphosphoinositol tetrakisphosphate (5,6[PP]2-InsP$_4$) and inositol hexakisphosphate (InsP$_6$). Our study also reveals that, in comparison to the eight typical human BRD families, the BRDL domain of PA200 has a shorter ZA loop and cannot be classified into any of those families.

## Results

### Cryo-EM analysis of the human PA200 and PA200-20S proteasome

Recombinant PA200 (approximately 2 mg/l of infected culture) containing N-terminal histidine tag was expressed in insect cells and purified by nickel affinity chromatography and gel filtration. Initially, we tried to obtain homogeneous PA200-20S and PA200$_2$-20S by incubating excess PA200 with commercially available 20S CP (50 μg/package, Boston Biochem, lot. #16518518). However, the 20S CP was the limiting factor in obtaining sufficient amounts of incubated sample, which made it difficult to separate the PA200-20S and PA200$_2$-20S complexes by gel filtration. Nevertheless, thanks to the advances in cryo-EM, 3 μg of the heterogeneous sample was sufficient to carry out structural data collection and pick out the particles that interested us for structure determination.

The PA200-20S complexes were obtained by incubating PA200 with human 20S CP at molar ratios of 2.2:1, 4.4:1, and 8.8:1 and then analyzed by negative staining EM. Complex formation at the molar ratio of 2.2:1 resulted in a mixture of single-capped PA200-20S complexes and uncapped 20S particles. At molar ratios of 4.4:1 and 8.8:1, 33.3% (4,119 out of 12,363) and 47.6% (7,642 out of 16,049) of the complexes were double-capped PA200$_2$-20S, whereas 51.0% (6,305 out of 12,363) and 40.2% (6,455 out of 16,409) of the complexes were single-capped PA200-20S, respectively (**Fig 1A and 1B**). Thus, the data show significant heterogeneity of

PA200 binding with the 20S regarding the assembly of single- or double-capped complexes, even when the concentration of PA200 was more than four times higher than that of 20S in our experiment.

In order to perform high-resolution structural analysis and direct comparison of α-subunit rings rearrangement with and without bound PA200, we applied cryo-EM sample preparation and imaged the PA200 monomer and single-capped PA200-20S proteasome complex using a Gatan K2 direct electron detector mounted on an FEI Titan Krios electron microscope. We picked 220,000 and 180,000 particles from 526 monomers and 1,300 complex micrographs, respectively (**Fig 1C and 1G**). Following reference-free 2D classification of PA200 and PA200-20S complex, we performed autorefinement on 103,000 PA200 particles and 87,000 single-capped PA200-20S complex particles (**Fig 1E and 1I**). Finally, we obtained the PA200 map with the resolution of 3.75 Å (EMD-0780 and PDB: 6KWX) and refined the resolution of PA200-20S complex to 2.72 Å (EMD-0781 and PDB: 6KWY) via block-based reconstruction by using the threshold value of 0.143 for the gold-standard Fourier shell correlation (FSC) (**Fig 1D, 1F, 1H and 1J**, **S1 Fig** and **S1 Table**).

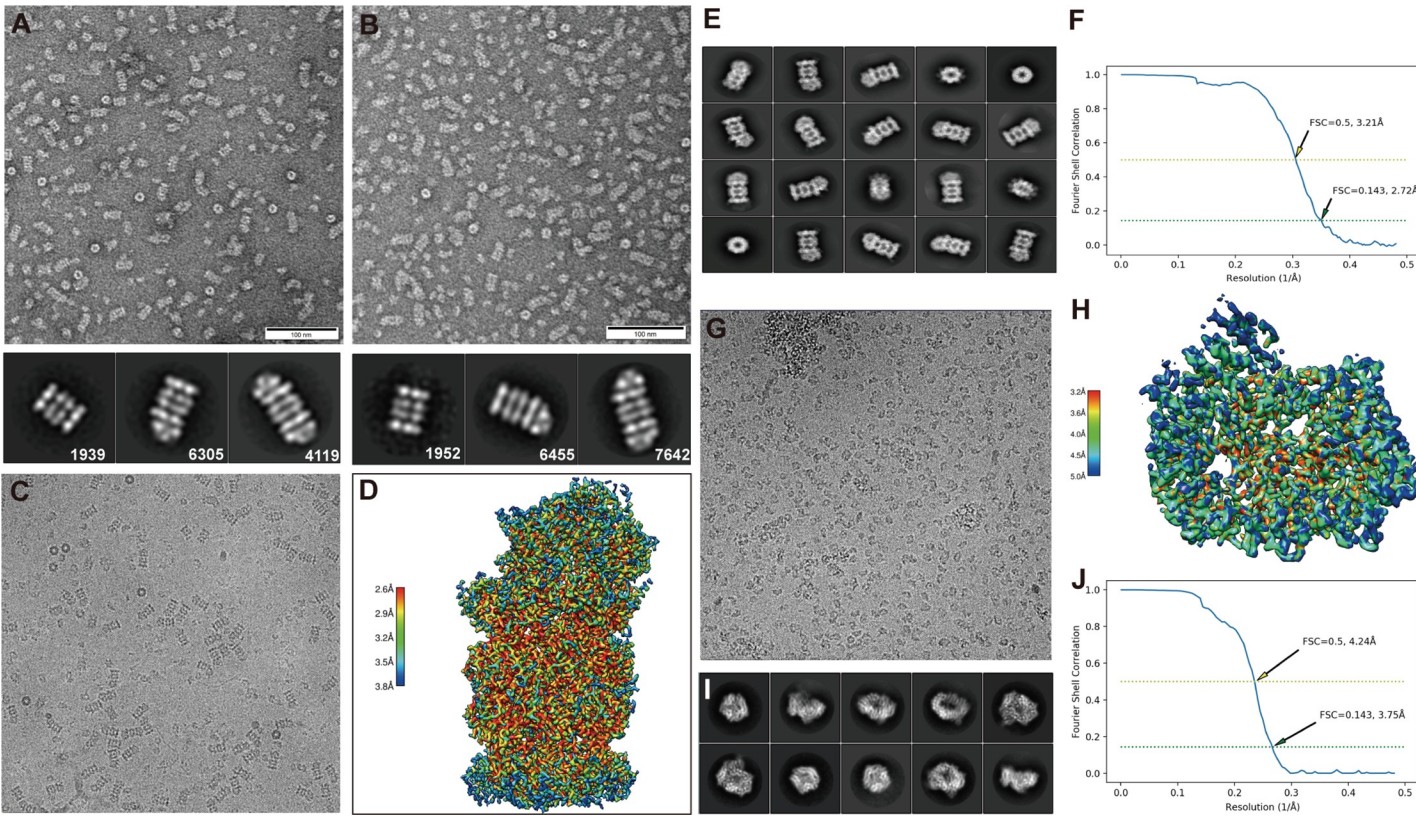

**Fig 1. NS-TEM images and cryo-EM structures of human PA200 and PA200-20S proteasome.** (A) PA200 and 20S were incubated at the ratio of 4.4:1 at room temperature for 1 h and then imaged by NS-TEM. Different particles were classified by 2D class averages and counted. Of the complexes, 33.3% (4,119 out of 12,363) were PA200$_2$-20S (double-capped), and 51.0% (6,305 out of 12,363) of the complexes were PA200-20S (single-capped). (B) Composition of three different particles at the PA200:20S ratio of 8.8:1. PA200$_2$-20S and PA200-20S were 47.6% (7,642 out of 16,049) and 40.2% (6,455 out of 16,409) in the complex. (C) Representative micrographs of frozen-hydrated PA200-20S complex particles. (D) Final cryo-EM density map of PA200-20S colored according to the local resolution. (E) Representative 2D class averages in different orientations of the complex. (F) Gold-standard FSC curves of PA200-20S complex. The resolution was determined to be 2.72 Å, and the 0.5 cutoff value is indicated by a horizontal yellow dashed line. (G) Representative micrographs of frozen-hydrated PA200 particles. (H) Final cryo-EM map of PA200 colored according to the local resolution. (I) Representative 2D class averages in different orientations of PA200. (J) Gold-standard FSC curves of PA200. The resolution was determined to be 3.75 Å, and the 0.5 cutoff value is indicated by a horizontal yellow dashed line. cryo-EM, cryo–electron microscopy; FSC, Fourier shell correlation; NS-TEM, negative-stain transmission electron microscopy; PA200, proteasome activator 200.

## Overall structures of the human PA200 and PA200-20S complex

Human PA200 contains 1,844 residues and has a theoretical molecular weight of 200 kDa, which makes it smaller than its yeast ortholog Blm10 (250 kDa) [21]. PA200 is shaped as a dome-like structure composed almost entirely of α-turn-α modules known as 32 HEAT repeat (HR)-like modules (**Fig 2A and 2B**, **S2 Fig**). The first ordered residue is Arg24, which is followed by a short loop before starting HR1 at Asp45. The other 31 HRs continue almost all the way to the C terminus and spiral through a 1.5-turn left-handed solenoid resembling a snail shell. Although the protein is predominantly composed of multiple HR modules, there are additional structures, such as independent helices α33 and α34 between HR16 and HR17, α41 between HR19 and HR20, α46 between HR21 and HR22, and α63 between HR29 and HR30. Moreover, three long loops interconnect HRs, the largest one being loop 1 (89 residues long) located between residues 1,215 and 1,303. Loop 2 and loop 3 are located between residues 729 and 773 and between 827 and 865, respectively (**S2 Fig**). Despite significant difference between primary structures of PA200 and Blm10, both proteins exhibit the same tertiary structure, which assumes a 1.5-turn left-handed solenoid shape [21]. The structural comparison between these two proteins (both from the complex) gave a root-mean-square deviation (RMSD) of 2.94 Å with 1,423 aligned residues (**Fig 2C and 2D**). PA200 is about 50 kDa smaller than Blm10 because of 301 residues deleted at different stages of evolution. These deletions mainly occurred at four locations, i.e., the N-terminal loop before HR-1A, the loop between HR-1A and HR-1B, the α-helices between HR10 and HR11, and the helices between HR26 and HR27. The N-terminal of PA200 contains only about 37 residues before the HR-1A that are involved in the formation of the opening 1. On the other hand, the N-terminal of Blm10 contains 125 residues with unknown function. Furthermore, the loop between HR-1A and HR-1B of PA200 contains only six residues, whereas its counterpart in Blm10 is composed of 86 residues (154–239) and is found adjacent to the largest opening on Blm10 [21].

PA200 can interact with 20S CP to assemble into three different proteasome types—PA200$_2$-20S, PA200-20S, and PA200-20S-19S—with each type present at distinct proportions in different organs and tissues [15,18]. The single-capped PA200-20S complex presented in our study is an asymmetric proteasome with one PA200 associated with one end of the 20S (**Fig 2E**). Though PA200 can interact with all seven α subunits of the 20S, the contact between the two molecules is asymmetrical, with a lateral opening situated over the α1, α6, and α7 subunits because of the 1.5-turn left-handed spiral structure of PA200 (**Fig 2F**, **S7B Fig**). Previously, a 23 Å–resolution EM study suggested that only six α subunits interact with PA200 and α7 subunit does not [25]. However, with the assistance of our high-resolution structures, we found that the α subunits 1, 6, and 7 have a smaller contact area than the other four subunits, and the α7 interacts with PA200 through its N-terminal loop after the conformational change induced by combining with PA200 (**Fig 2F**).

Although the resolution of PA200 was much lower than that of PA200-20S complex, the overall comparison of PA200 and the PA200-20S complex gave an RMSD of 0.78 Å with 1,719 aligned residues. Some regions (i.e., residues 562–575, 1,809–1,843) vanished in PA200 but were evident in the PA200-20S complex, suggesting that these flexible regions tend to be stabilized by intermolecular interactions (**Fig 2G and 2H**).

## PA200 binding induces gate opening in the 20S CP

Upon binding of PA200, the central region of the α-rings in the 20S CP is rearranged to yield an apparent open channel conformation expected to allow passage to small molecule substrates rather than proteins (**Fig 3A and 3B**). In addition, the N-terminals of α subunits at the PA200-interacting end of the 20S underwent a substantial conformational change. The α1–α4

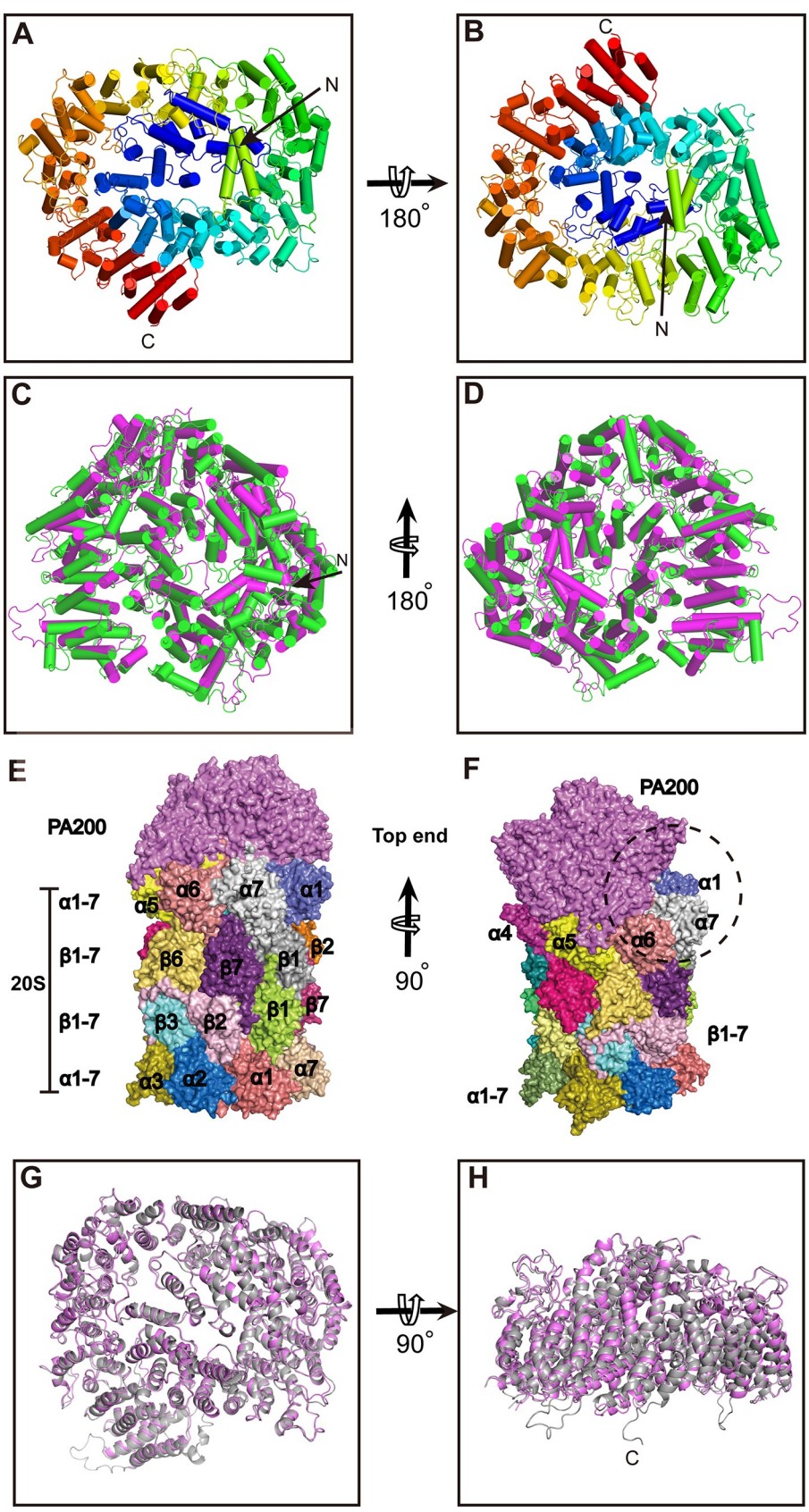

**Fig 2. The overall structures of PA200 and PA200-20S.** (A) PA200 (cylindrical helices) forms a largely closed dome-like shape as seen from the top view. The N-terminal is indicated with a black arrow. (B) Same as (A) but the dome is rotated 180˚ to show the bottom view. (C) Structural alignment of PA200 (PDB: 6KWY) and Blm10 (PDB: 4V7O), shown as top view. Blm10 (green) is slightly bigger than the dome of PA200 (magenta). (D) Bottom view of the structural alignment of PA200 and Blm10. (E) Side view, space-filling representation of PA200-20S complex. PA200 is shown as magenta, and the subunits of the 20S are shown in different colors. (F) Same as (E) but the complex is rotated 90˚ and with a lateral chip from this perspective. The top subunits α1, α6, and α7 have smaller volume interaction with PA200. (G) The overall structure of PA200 (gray) from the PA200-20S complex structurally aligned with unbound PA200 (magenta). (H) Same as (G) but the dome is rotated 90˚ to show the side view. Blm10, Bleomycin resistance 10; PA200, proteasome activator 200; PDB, Protein Data Bank.

subunits became disordered, which respectively resulted in the absence of 8, 17, 30, and 18 residues of α1–α4 in the density map and led to opening of the 20S gate. In contrast, only N-terminals of α2–α4 subunits of Blm10 are missing in the Blm10-20S complex [21]. Additionally, the complete N-terminals of α5–α7 vertically insert into the PA200. The unbound end of the 20S in our complex has the same conformation as the uncapped 20S (**Fig 3C and 3D**, **S3 Fig**). Our 3.3-Å uncapped 20S map showed that both gates are closed by the N-terminal tails of α subunits prior to binding PA200 (**S3 Fig**).

Although the two ends of 20S are identical, we henceforth refer to the end that binds PA200 in our PA200-20S complex structure as the top end. We also compared the 20S from our complex with the human 20S reported in the previous study (PDB: 5LEX) [26]. Whereas the bottom α subunits are mostly unchanged (RMSD = 0.62 Å over 1,657 Cα atoms), the top α subunits of our complex display a slight conformational change in comparison to the other reported structure (RMSD = 1.57 Å over 1,497 Cα atoms) and form a disordered gate. Two different slice forms of the 20S-PA200 complex density map give a clear view of the opened top gate induced by PA200 binding and the closed bottom gate (**Fig 3E and 3F**).

To validate the impact of PA200 on the enzymatic activity of the proteasome, we examined the chymotryptic-like activity of the 20S and 20S-PA200 complex. To this end, we used a chymotryptic-like fluorogenic peptide Suc-LLVY-AMC that can induce the gate opening and be degraded by both 20S CP and the 20S-PA200 complex, albeit at higher rates by the complex [17]. Indeed, our results confirm that 20S cleaves the Suc-LLVY-AMC with the $V_{max}$ of 3.14 μM/min, whereas binding of PA200 enhances the peptidase activity to 11.37 μM/min (**Fig 3G**).

PA200 makes numerous contacts with the α-ring surface of 20S by using the C-terminal hydrophobic-tyrosine-other (HbYX) motif and a loop (T562 to K574) to contact the N-terminals of 20S subunits α5, α6 and α1, α2 (**Fig 4A–4F**). The HbYX motif of PA200 (YYA) inserts into the pocket between proteasome subunits α5 and α6 in a manner similar to Blm10, 11S activator (PA26 and PA28), and 19S/PAN activators [14,21]. These interactions play a crucial role in displacing the Pro17 turn and subsequent opening of the proteasome gate. The side chains of Tyr1841 and Tyr1842 form a hydrogen bond with the oxygen atom of α5 Glu25 and α5 Gly19, respectively. The interaction between Tyr1842 and α5 Gly19 stabilizes the adjacent α5 Pro17 reverse turn. In addition, the interaction between Tyr1841 and α5 Glu25 further assists in holding the C-terminal of the α5 subunit (**Fig 4B**).

## Two openings on PA200 bind inositol phosphate cofactors

HRs in PA200 wrap into a solenoid-like dome that caps the 20S surface, covering the proteasome entrance pore and positioning PA200 to regulate traffic into the 20S. PA200/Blm10 stimulates the hydrolysis of small peptides or misfolded proteins during the maintenance of mitochondrial functions [21]. The complex structure of Blm10-20S shows a big lateral opening (13 × 22 Å) located at the dome-like structure of Blm10 (**Fig 5A**). In contrast, the PAN

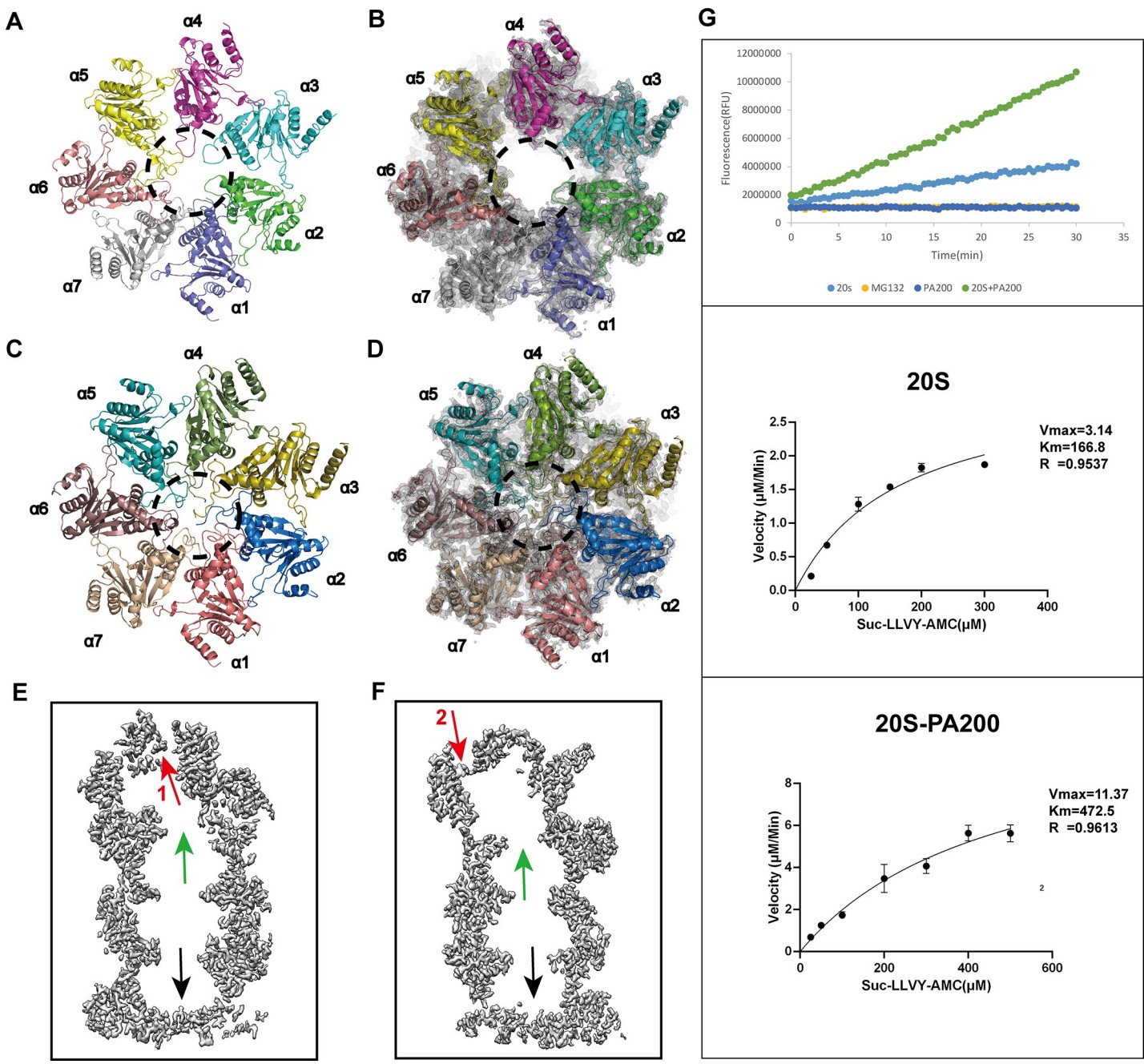

**Fig 3. Gate opening and the peptidase activity of the 20S induced by PA200.** (A) After the combination of PA200, the gate of the 20S formed by top α subunits became disordered (shown within a dotted circle). (B) Close-up views of the top α-rings. The cryo-EM maps are shown as gray mesh and the atomic models as a cartoon. The N-terminals of the top α subunits became disorder and invisible. (C) The bottom gate of the 20S is closed because PA200 is not bound to it to induce gate opening. The N-terminals of all seven α-subunits point to the center of the hole and close the gate. (D) Close-up views of the bottom α-rings. The cryo-EM map (gray mesh) with a fitted atomic model (cartoon representation). Unlike panel B, the N-terminals of the bottom α subunits point to the central of the closed gate. (E) A surface cut of density map gives a clear sectional view with the open gate (green arrow) on top, closed bottom gate (black arrow), and opening 1 with the density map of the cofactor (red arrow). (F) The other cutting angle of the sectional view gives further proof of the two gates and the opening 2 with its cofactor map. (G) The proteasome activity of the 20S and PA200-20S was evaluated by the 20S proteasome assay kit, and Suc-LLVY-AMC was used as substrate. First, 20S CP and PA200-20S CP (2.5 nM) were incubated with 100 μM Suc-LLVY-AMC for 15 min at 25˚C, and then fluorescence measurements (RFU) were taken at 30-s intervals and plotted against time. Data underlying these plots for top panel can be found in **S1 Data**. The chymotryptic-like activity of 20S and 20S-PA200 were analyzed at different concentrations of Suc-LLVY-AMC (25, 50, 100, 150, 200, 300 μM for 20S and 25, 50, 100, 200, 300, 400, 500 μM for 20S-PA200). The underlying numerical data and statistical analysis for middle and bottom panels can be found in **S2 and S3 Data**. CP, core particle; cryo-EM, cryo–electron microscopy; PA200, proteasome activator 200; RFU, relative fluorescence units.

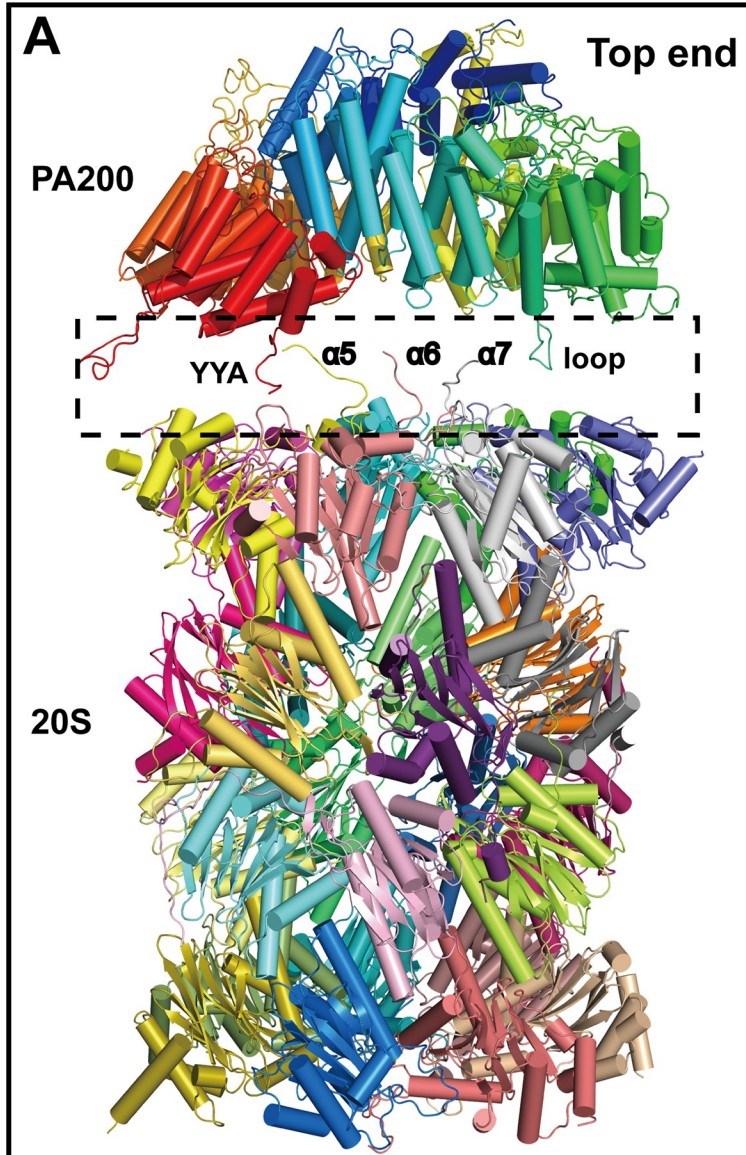

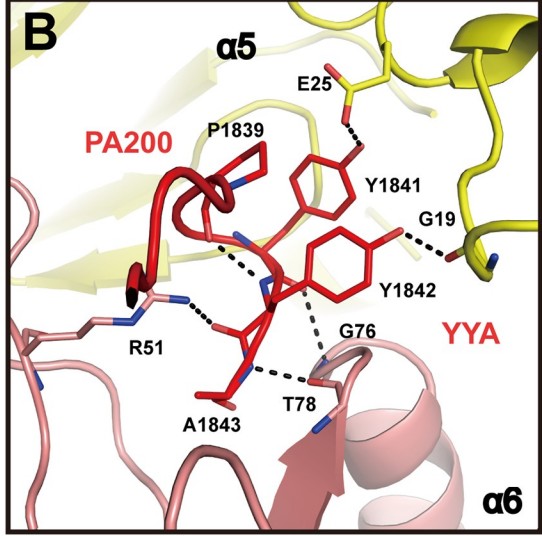

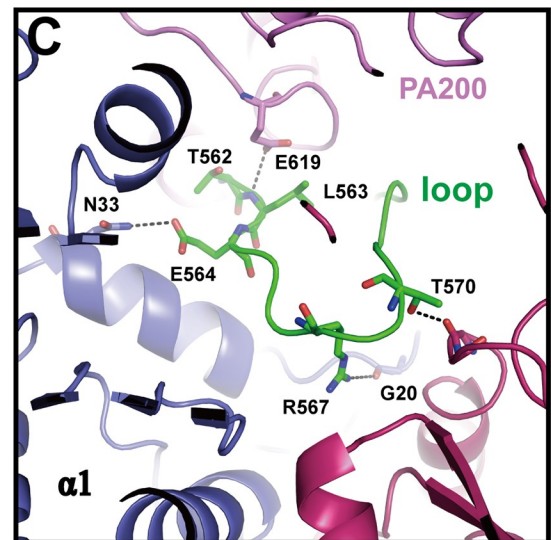

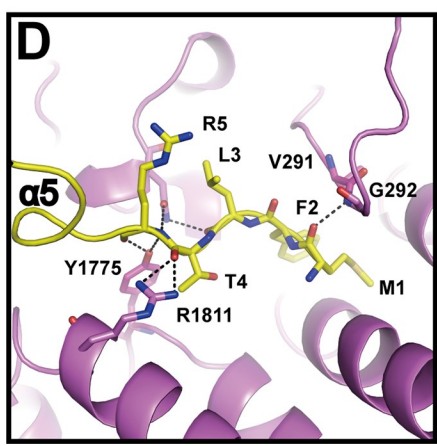

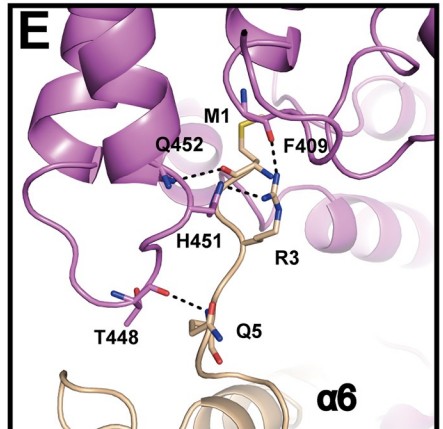

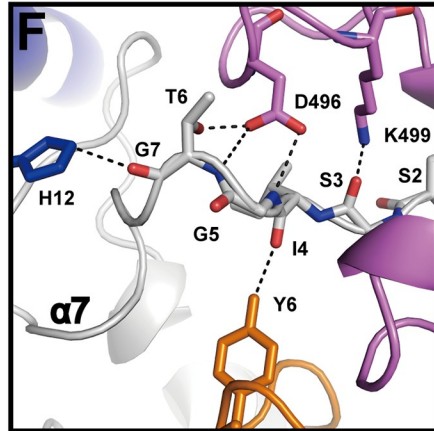

**Fig 4. Interactions between PA200 and 20S.** (A) Five important parts involved in the interactions are shown in the black dashed box after a little movement of PA200 apart from the 20S. (B-F) Close-up views of contact areas YYA (dark green) (B), loop (562–572, green) (C), α5 (N-terminal of subunit α5, lemon) (D), α6 (N-terminal of subunit α6, cyan) (E), and α7 (N-terminal of subunit α7, gray) (F). Key residues involved in interactions between PA200 and 20S are labeled and shown as sticks. The hydrogen bonds are shown as dashed lines. PA200, proteasome activator 200.

complex from *Methanocaldococcus jannaschii* has an opening with a diameter of 13 Å [20,27]. However, the electron densities of the amino acid (aa) residues outlining the mouth of this opening in the Blm10 structure—i.e., residues Leu154-Asn239 and Tyr1037-Leu1147—are not visible. In our cryo-EM structures of PA200 and PA200-20S complex, three loops cover this putative opening: loop 3 (827–865, colored blue), loop 4 (953–960, colored red), and loop 5 (993–999). Loop 3 forms a rectangular lid covering this region, whereas loop 4 shifts toward the center of the putative opening (**Fig 5B**, **S7C Fig**). Moreover, the two loops also form numerous interactions with other regions of PA200 to lock the lid on the hole. Five hydrogen bonds between loop 3 and HR1 (1A, 1B, and loop 5) and one hydrogen bond between Glu834 and Lys62 help to fix the big loop 3 on the putative opening. The loop 4, slanting through the middle of the opening, forms two hydrogen bonds, between Glu956 and HR1B-Thr81, Gly957, and Asn998. Therefore, this part does not seem to be an opening in PA200 (**Fig 5A and 5B**).

Surprisingly, there are other two pores with prominent openings (henceforth referred to as openings 1 and 2) located on PA200 (**Fig 5C**). One is sitting at the top of PA200 with the size of $19.6 \times 13.7$ Å when measured between Cα atoms, whereas the other is sitting at the side of PA200 with a size of $23.3 \times 17.8$ Å when measured between Cα atoms (**Fig 5D**). Even more surprising is that the side chains of several positively charged residues outlining the two openings point toward the center of the respective openings, where they bind notably well-defined densities. Based on the unique shape of these densities and the function of PA200, we speculated that they are inositol phosphate moieties [28]. We thus modeled inositol phosphates into the density map and refined them. The refinement indicated that the density map bound by opening 2 corresponded to $InsP_6$, whereas the density map bound by opening 1 was identified as $5,6[PP]2\text{-}InsP_4$ ($C_6H_{20}O_{30}P_8$) because of extra densities at two adjacent groups. To validate our structural data, standard $InsP_6$ (phytic acid sodium salt hydrate, $C_6H_{18}O_{24}P_6 \cdot xNa^+ \cdot yH_2O$, MERCK, P8810) and purified PA200 (20 mM HEPES [pH 7.5] and 150 mM NaCl, 1 mM Dithiothreitol [DTT]) were boiled for 5 min at 95°C and analyzed by high-performance liquid chromatography–mass spectrometry (HPLC-MS). The standard $InsP_6$ produced a characteristic peak at 814.85 $(M_{InsP6}+7Na)^+$ (**Fig 6A**), and PA200 produced several characteristic peaks. In addition to the characteristic peak of 814.85 $(M_{InsP6}+7Na)^+$ in PA200, several other peaks representing different forms of sodium $5,6[PP]2\text{-}InsP_4$ with varying number of $Na^+$ ions were observed, such as peak 1,019.19 $(M_{5,6[PP]2\text{-}InsP4}+9Na)^+$, peak 1,041.13 $(M_{5,6[PP]2\text{-}InsP4}+10Na)_+$, and peak 1,063.24 $(M_{5,6[PP]2\text{-}InsP4}+11Na)^+$ (**Fig 6B**). On the basis of all evidence, we conclude that the two small molecules bound by the two openings on PA200 are $5,6[PP]2\text{-}InsP_4$ and $InsP_6$ (**Fig 5D**, **S4 Fig**, **S7D and S7E Fig**).

## The noncanonical BRDL domain of PA200 recognizes AC histones

Lysine acetylation serves as a marker on the core histones and is directly recognized by PA200 at the loci of DSBs [18]. The recognition of AC histones is generally performed by BRDs that generally employ asparagine residues to bind the Kac on core histones. BRD folds into an evolutionary conserved structure of approximately 120 residues that comprises a left-handed bundle of four α-helices interconnected with three loops in which two α-helices are adjacent to hydrophobic ZA and BC loops (**S6 Fig**). Thus far, 61 BRDs located in 46 diverse proteins and clustered into eight distinct families have been identified in the human genome [24].

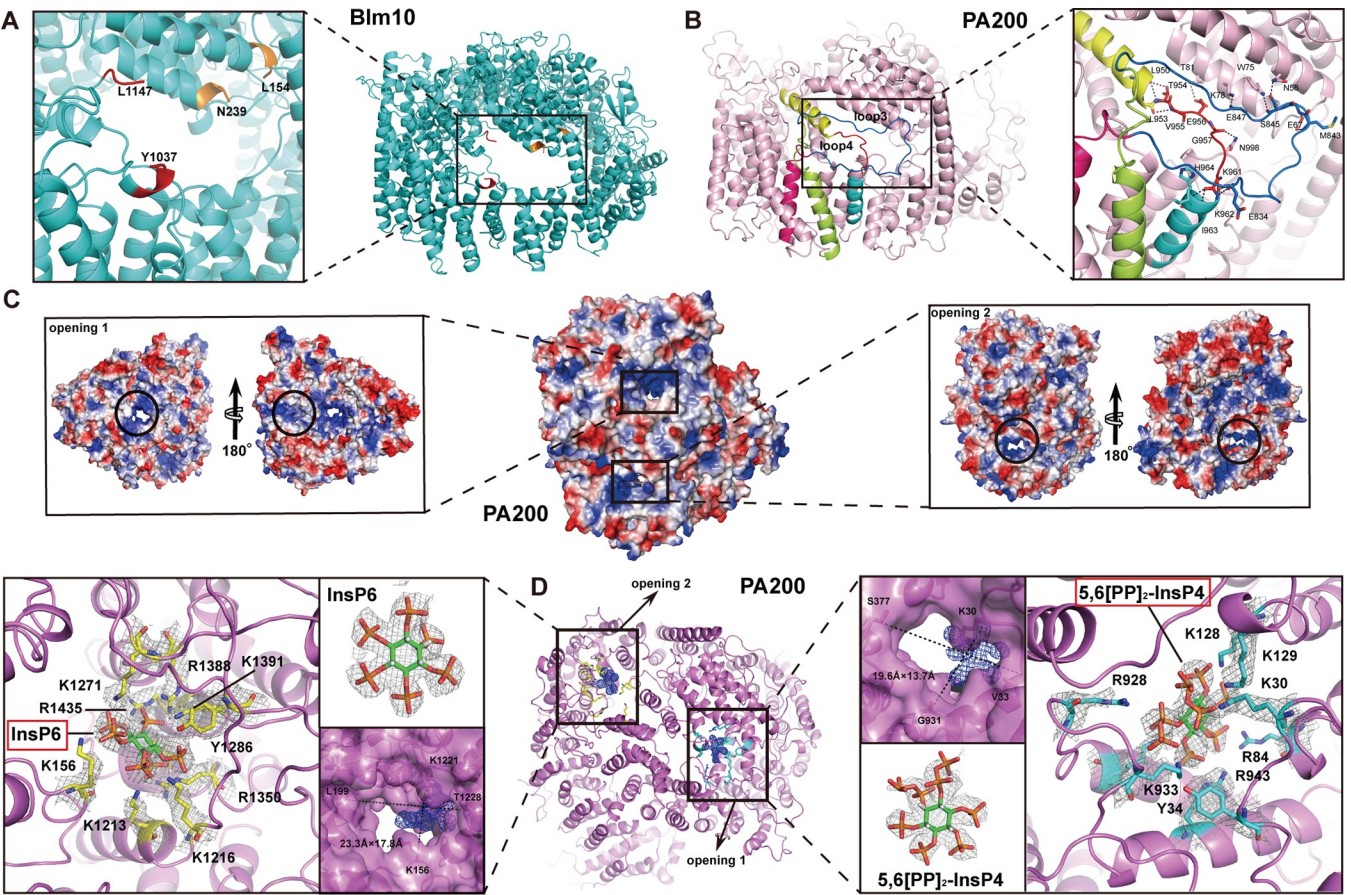

**Fig 5. Structural details of the two openings and the cofactors bound by PA200.** (A-B) Close-up view of the openings on Blm10 (PDB: 4V7O) (A) and PA200 (B) dome. The location on PA200 corresponding to the largest opening of Blm10 is covered by two loops (loop3, blue; and loop4, red). Residues on the two loops involved in the interaction with other nearby residues are shown as sticks, and the hydrogen bonds are shown as dashed lines. Two regions are missing in the previously reported Blm10 structure (L154–N239 and Y1037–L1147). However, these two regions happen to be the reported Blm10 opening. (C) PA200 is shown as electrostatic surface, and the two openings are shown in black circles. (D) Close-up views of the two openings on PA200 and its cofactors. Positively charged residues located at the openings and involved in the interaction are shown as sticks indicated in different colors (cyan sticks in opening 1 and yellow sticks in opening 2). The electron densities bound in opening 1 and 2 are compatible with $5,6[PP]_2$-$InsP_4$ and $InsP_6$, respectively. The dimensions of the two openings are $19.6 \times 13.7$ Å and $23.3 \times 17.8$ Å, respectively. $5,6[PP]2$-$InsP_4$, (5,6)-bisdiphosphoinositol tetrakisphosphate; Blm10, Bleomycin resistance 10; $InsP_6$, inositol hexakisphosphate; PA200, proteasome activator 200.

In the human PA200, although the 82-residue BRDL domain folds into the same geometry as typical human BRDs, it does not share significant sequence homology with them [20]. BRDL domain of PA200 forms four α helices (αZ, αA, αB, and αC) and three loops (ZA loop, AB loop, and BC loop). Unlike the typical BRD that contains a 24–37 aa insertion loop with one or two short α-helices, the ZA loop of PA200 is only 9 residues long and contains a 5-residue small helix, whereas that of Blm10 is even shorter and comprises only 5 residues (**Fig 7A and 7B**, **S6A** and **S7F** Figs) [18]. Analogously to other known BRDs, PA200 has been reported to bind AC core histones with two key asparagine and phenylalanine residues located on BC loop, i.e., Asn1716 and Phe1717 [18]. By contrast, the BRDL domain of Blm10 recognizes Kac with residues Tyr1663 and Asn1664 located on the ZA loop (**S5A and S5B Fig**) [18].

Altogether, our data indicate that both primary and ternary structure of the PA200 BRDL domain are markedly different from the eight canonical BRD families and could not be classified into any of them (**S5 Fig**).

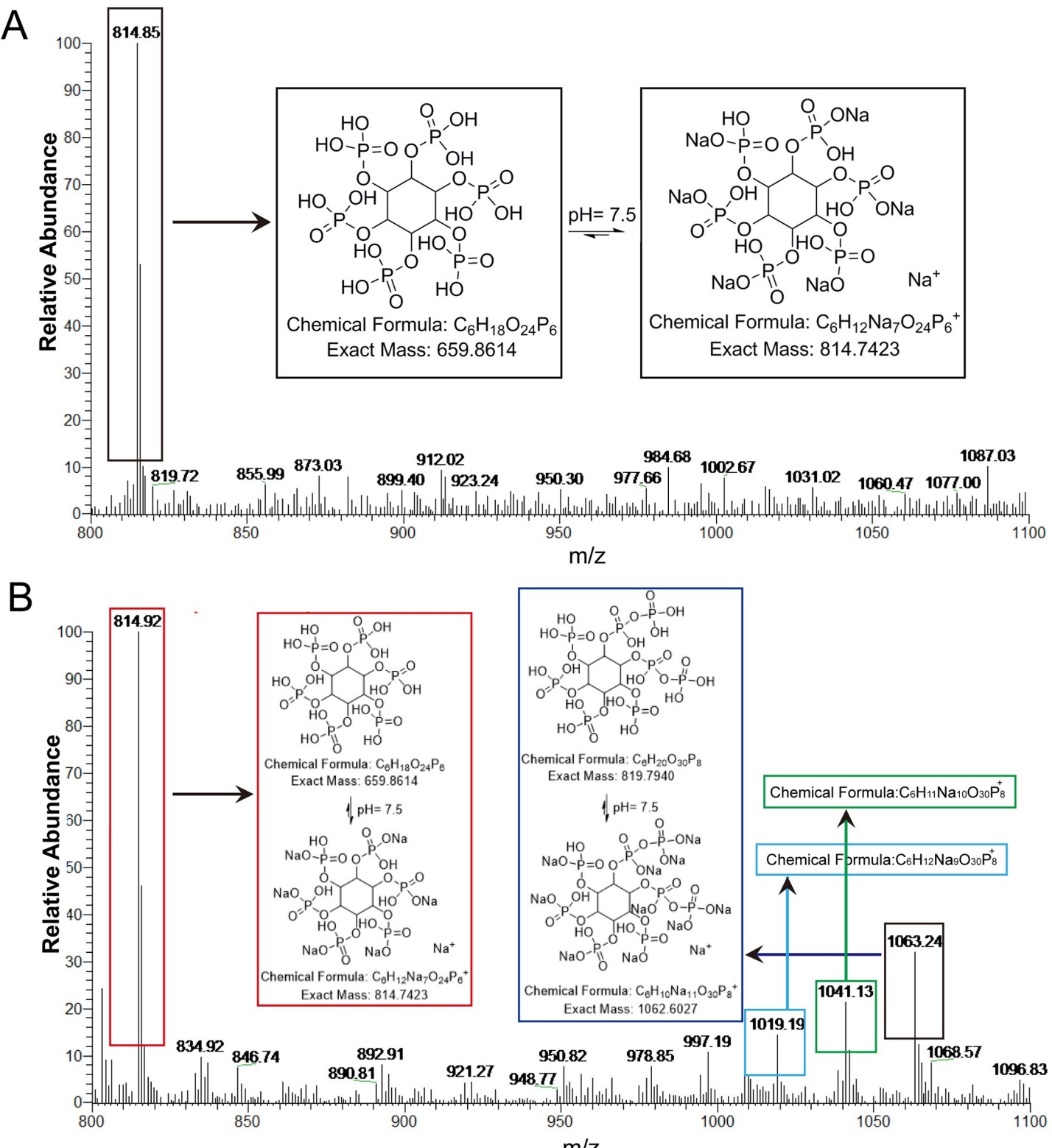

**Fig 6. Identification of 5,6[PP]₂-InsP₄ and InsP₆ by HPLC-MS.** (A) The InsP₆ standard (phytic acid sodium salt hydrate, $C_6H_{18}O_{24}P_6 \cdot xNa^+ \cdot yH_2O$) was analyzed by HPLC-MS system. The main characteristic MS peak at 814.85, corresponding to $(M_{InsP6}+7Na)^+$. The molar mass of InsP₆ is 660.029 g·mol⁻¹. (B) PA200 was validated using the same program, and the MS gave one characteristic peak at 814.92 corresponding to $(M_{InsP6}+7Na)^+$ (red) and a series of peaks that represent different forms of sodium 5,6[PP]2-InsP₄ with various number of Na⁺ ions, i.e., peak 1,019.19 $(M_{5,6[PP]2-InsP4}+9Na)^+$ (cyan), peak 1,041.13 $(M_{5,6[PP]2-InsP4}+10Na)^+$ (green), and peak

1,063.24 ($M_{5,6[PP]2\text{-}InsP_4}$+11Na)$^+$ (blue). 5,6[PP]2-InsP$_4$, (5,6)-bisdiphosphoinositol tetrakisphosphate; HPLC-MS, high-performance liquid chromatography–mass spectrometry; InsP$_6$, inositol hexakisphosphate; PA200, proteasome activator 200.

## Discussion

PAs play essential roles in the regulation of proteasome activity by instigating the 20S gate opening, thereby permitting translocation of substrate proteins into the proteolytic cavity. Non-ATPase activators PA200/Blm10 can bind with the 20S and stimulate protein degradation under certain situations such as metabolic adaptation and stress response rather than stimulate common proteolysis of intact globular proteins [29–31]. Our structures revealed that PA200 is shaped as a dome-like structure and consists of HR-like modules [32], each including two helices linked by a turn, with neighboring repeats associated by a linker (**Fig 2A and 2B, S2 Fig**). Moreover, we provide evidence that PA200 binds with activated 20S CP to form both symmetric and asymmetric varieties (single-capped PA200-CP and double-capped PA200$_2$-CP) (**Fig 1A and 1B**) [33–35]. The cryo-EM structure of PA200-20S complex in this study demonstrates binding of PA200 to the outer-ring surface of the 20S cylinder, thus achieving the topological requirements for a PA (**Fig 2E and 2F**).

The previous studies have shown that uncapped, single-capped, and double-capped proteasomes coexist both in vivo and in vitro conditions. However, the physiological function of such complex heterogeneity is still poorly understood [18]. In our study, we were unable to obtain homogeneous samples of either PA200-20S or PA200$_2$-20S at any tested molar ratio between PA200 and 20S, even when it was as high as 8.8:1. We believe that two possible reasons prevented us from obtaining homogeneous complex samples: (1) Saturation of 20S capping cannot be achieved by merely increasing the PA200 concentration. Under physiological conditions, proteasomes in eukaryotes, archaea, and some bacteria need to bind proteasomal activators at either one or both ends of the proteasome CP for full activity [36]. Hybrid proteasomes with different activators on either end of the CP cylinder have also been observed [37]. Moreover, more than 50% of the cellular 20S proteasomes exist in an activator-unbound form [16]. With regard to PA200, our findings are supported by the recently published article in which the complexes were prepared by coexpressing the two proteins in insect cells [38]. In spite of different experimental procedure for protein expression and purification, their cryo-EM images also showed varying proportions of uncapped, single-capped, and double-capped complex particles, with 28% of the particles in the final volume classified as single-capped complex [38]. Overall, the studies currently suggest that proteasomes indeed exist as a mixture of uncapped, single-capped, and double-capped complexes under physiological conditions, albeit for unknown reasons. (2) The freshness and binding efficiency of the protein sample may have been impacted by freeze-thaw cycles to a certain degree, particularly since the 20S protein sample used in complex preparation is an endogenously expressed commercial protein. However, this reason may not be as essential for the observed results as the first reason.

The 20S proteasome is responsible for the degradation of disordered and oxidized proteins [39], as well as natively unfolded proteins such as tau [40]. PA200 binding augments peptidase activity of 20S CP and assists traffic through the gated channel (**Fig 3G**). This traffic is noticeable in our cryo-EM structure of the PA200-20S complex, which shows that binding of PA200 initiates extensive rearrangements within the central-pore region of the α-ring into an open channel conformation (**Fig 3E and 3F**) [41]. The open channel conformation is formed by the orientation of PA200 monomer toward the binding sites on the central region of 20S α-ring [17,33,42,43]. Whereas PA200 induces the rearrangement of the central region of the top end α-rings into an apparent open channel conformation, the PA200-unbound end is still closed

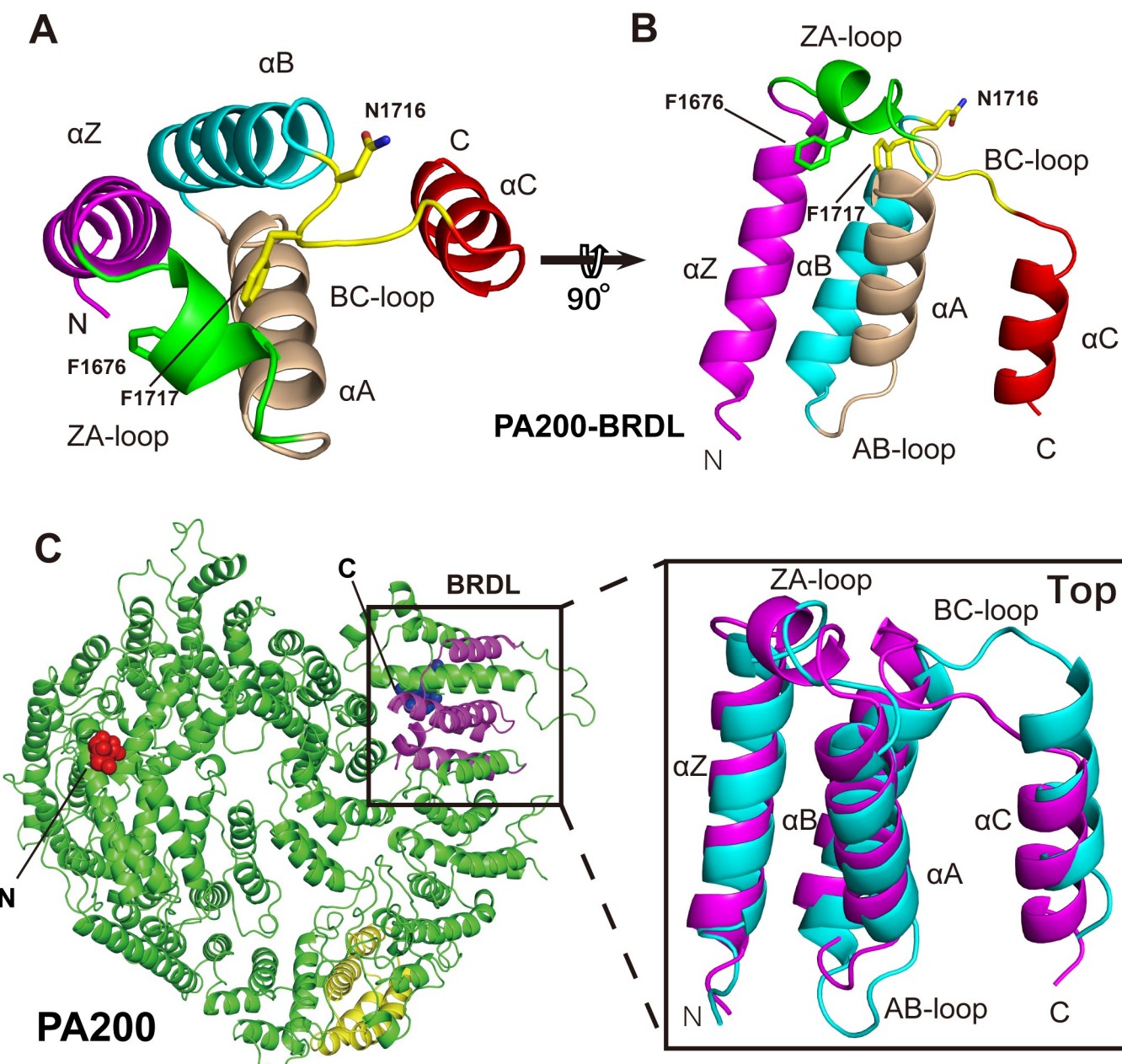

**Fig 7. The BRDL domain of PA200.** (A-B) Ribbon representation of the PA200 BRDL domain in top view (A) and side view (B), with the key residues shown as sticks. (C) The alignment of the PA200 BRDL domain (magenta) and the corresponding region (cyan) of Blm10. The region corresponding to BRDL domain of Blm10 is shown as yellow. The N-terminal of Blm10 is shown as red spheres, whereas the C-terminal is indicated as blue spheres (see also S4 Fig). Blm10, Bleomycin resistance 10; BRDL, bromodomain-like; PA200, proteasome activator 200.

(**Fig 3A–3D**, **S3 Fig**). The N-terminals of α5–α7 subunits of the 20S interact vertically with PA200, whereas the C-terminal (YYA) and a loop (Thr562 to Lys574) of PA200 insert into 20S (**Fig 4A–4C**). This interaction is energetically favorable for α-subunits to destabilize the closed conformation and permit the formation of an open conformation [43]. Our structural data show that the C-terminal (YYA) of PA200 plays a key role in interaction with the 20S by using side chains of Tyr1841 and Tyr1842 to form hydrogen bonds with α5 Gly19 and Glu25, thus stabilizing the adjacent α5 Pro17 reverse turn and displacing it to activate proteasome gate

opening [44–47] (**Fig 4B**). The penultimate tyrosine of PA200 establishes interactions identical to the penultimate tyrosine of Blm10 subunit, with the monomeric PA200 moving a single proteasome α-subunit Pro17 turn and inducing a conformational change into a fully open-gate state [48,49]. Hence, our study supports the model in which PA200/Blm10, 11S, and 19S/PAN all bind proteasome CPs via their C-terminal residues, which triggers partial or full opening of the proteasome gate by displacing single or multiple Pro17 turns. In order to initiate proteasome gate opening, PA200 and PAN/19S utilize a penultimate tyrosine/phenylalanine to displace the Pro17 turn, whereas PA26/11S uses an internal "activation loop" [43]. However, although PA200/Blm10, 11S, and 19S/PAN may open the gate with similar mechanism [1], it is unclear how monomeric PA200/Blm10 displaces one Pro17 turn from a single proteasome subunit by YYA to open the gate.

Besides the mechanism underlying PA200-mediated 20S CP gate opening, we also observed two new openings on PA200. These two openings are noticeably sized and outlined with positively charged residues (**Fig 5C**) [50]. The formation of the two openings in the PA200-20S structure is significantly different from that seen in the Blm10 equivalent. Moreover, our structures reveal three loops (loop 3, loop 4, and loop 5) and hydrogen bonds that cover and lock the putative region of the opening previously reported for the Blm10-20S complex (**Fig 5A and 5B**). Therefore, the functional significance of this opening may need further verification. In a recent study, PA28 from *Plasmodium falciparum* (PfPA28) was found to exhibit a heptameric structure with a charge-segregated pore formed by an approximately 20-Å positively charged apical (narrow) end and a 35-Å negatively charged basal end. The size of the pore (20 Å minimum) is sufficient for peptides and unfolded polypeptide substrates/products to pass through [14]. Given that the size of the two charged openings of PA200 (19.6 × 13.7 Å and 23.3 × 17.8 Å) in our study is similar to the pores of PfPA28 (**Fig 5D**) [14], it seems plausible that the two openings of PA200 are the gates for feeding substrates for proteolysis. Intriguingly, the positively charged residues of these two openings respectively bind 5,6[PP]2-InsP$_4$ and InsP$_6$ (**Fig 5D**, **S4 Fig**), which was confirmed by our HPLC-MS results (**Fig 6**). Previous studies have shown that D-myo-inositol-(1,4,5,6)-tetrakissphosphate (Ins(1,4,5,6)P$_4$) is a bona fide conserved regulator of class I histone deacetylases (HDACs) that acts as an "intermolecular glue" between HDAC3 and its corepressor silencing mediator of retinoid acid and thyroid hormone receptors (SMRT) [28,51,52]. Therefore, it is plausible to suggest that inositol phosphates play a role in histone degradation by PA200-20S proteasome, but this requires further validation [21].

Histone acetylation affects protein interfaces and chromatin approachability, thus playing a crucial role in transcriptional regulation, replication, and DNA repair [53]. Excess histones in a cell can trigger genome instability [54,55]. Hence, histones are acetylated and degraded in case of DNA damage [56]. Previously, it has been shown that PA200 plays a crucial role in the regulation of 20S-mediated proteolysis and enhancing the 20S degradation of AC core histones [57]. PA200 uses BRDL domain to specifically target AC core histones for proteasomal degradation, thus participating in acetylation-regulated histone degradation and DNA repair [18]. Our study shows that PA200 comprises the BRDL region analogous to the BRDL region of Blm10, with α-helices and hydrophobic residues in neighboring loops (**Fig 7**, **S5 Fig**). Although the three key residues of BRDL domain are indeed located in proximity to each other, the side chain of Phe1676 is buried in PA200 and the side chain of Asn1716 oriented away from the hydrophobic pocket of BRDL and points to a hydrophobic pocket formed by αB, αC, and two adjacent helices (**Fig 7A and 7B**) [38]. The Kac of histone is recognized and anchored in the central hydrophobic cavity by a hydrogen bond between Kac and a conserved asparagine residue in most BRDs, though relatively low affinities between Kac and BRDs suggest that some BRDs require more than one Kac site or posttranslational protein modifications

for efficient recognition of AC histones [58,59]. BRD4(1) is one of the human BRDs that can bind various peptides containing more than one Kac site, such as H4 $_{1-11}$ K5 ac K8 ac, H4 $_{11-21}$ K12 ac K16 ac, and H4 $_{15-25}$ K16 ac K20 ac, and even two BRD4(1) have been reported to bind with one H4 $_{7-17}$ K8 ac K12 ac peptide [24]. In contrast, K16ac is not sufficient for PA200 BRDL-mediated recognition of AC histones [18], implying that either more than one Kac site is required for extensive hydrogen bonding or additional posttranslational modifications of histones are necessary, or both. Based on the structure and multiple sequence alignments with other BRD family members, variations in critical residues and structural motifs suggest that the PA200 BRDL region does not belong to any of the currently recognized eight BRD families (**S5 Fig**).

Interestingly, although PA200 and Blm10 are homologous proteins, there are notable differences in the location of the BRDL regions and key Kac-binding residues between the two proteins. As seen in the **Fig 7C**, the key residues of PA200 BRDL are located on the top of the dome, whereas the key residues of Blm10 BRDL are located on the bottom of the dome, on the border between the Blm10 and 20S (**S5C Fig**). Both primary sequence and structural alignment of PA200 and Blm10 show that their BRDLs—aa 1,650–1,731 in PA200 (**Fig 7C, magenta**) and aa 1,650–1,732 in Blm10 (**S5C Fig, red**)—could not be aligned with each other. Instead, the BRDL of PA200 corresponds to the aa 1,961–2,042 region of Blm10 (**S5C Fig, cyan**), whereas the aa 1,356–1,432 region (**Fig 7C, yellow**) of PA200 corresponds to the BRDL of Blm10. The structural alignment of the PA200 BRDL with the cyan region of Blm10 gave the RMSD of 1.32 Å (70 Cα atoms), whereas the structural alignment of the Blm10 BRDL with the yellow region of PA200 gave an RMSD of 2.11 Å (78 Cα atoms). In spite of the seeming correspondence of structure alignment, the yellow region of PA200 and cyan region of Blm10 do not share any sequence homology with BRD domains and possess neither a typical motif nor key residues for reading Kac, so we conclude that they are not true counterparts to the BRDLs of PA200 and Blm10 and are thus unlikely to mediate recognition of AC histones. This unexpected alignment between BRDL of PA200 with the cyan region of Blm10 and BRDL of Blm10 with the yellow region of PA200 may be due to a relatively large difference in size between PA200 and Blm10 (approximately 50 kDa) and/or some other unknown reasons. Moreover, these alignments, the fact that there is certain degree of fold similarity between them (as seen from RMSD values), and overall protein structure abundant in HRs could raise the question of whether PA200 and Blm10 contain a Kac-binding BRDL domain and one or more BRDL-reminiscent folds. However, this notion and its functional implications need further investigation.

During our manuscript submission process, another structural study on PA200 and PA200-20S proteasome complex by Toste Rêgo and da Fonseca was published in *Molecular Cell* [38]. The study is highlighted by the successful assembly of the eukaryotic proteasome by recombinant expression and structures of uncapped 20S and double-capped PA200$_2$-20S complex. Moreover, the article discusses the allosteric modulation of 20S proteasome active sites by PA200. Similarly to our study, the authors also identify the two openings on PA200 and observe that they bind inositol phosphate cofactors. Although the scopes and results of the two studies partially overlap, the data are complementary and serve to support the findings of either study and are described and analyzed from two different perspectives to give a better all-around picture of the subject.

Altogether, our structures provide a solid basis for follow-up studies on proteolytic substrate (and product) traffic through the two openings on the PA200 dome into (and out of) the 20S CP and PA200 proteasome-mediated degradation of AC histones. In addition, the results of our study will not only give new insights for identifying other Kac-binding proteins with BRDL regions but will also be useful for development of new anticancer, anti-inflammatory,

and contraceptive drugs [60] that act by obstructing proteasomes essential for protein degradation in DNA repair.

## Methods

### Purification of human PA200

Human PA200 was expressed in insect cells using the Bac-to-Bac expression system (Invitrogen). The cDNA corresponding to residues 1–1,843 of PA200 was cloned into the baculovirus transfer vector pFastBac1 (Invitrogen). Transfection and virus amplification were carried out according to the regular protocol for baculovirus expression system (Invitrogen). Suspension cultures of *Trichoplusia ni* (Hi5) cells (Invitrogen) were cultured in insect SIM HF cell medium (Sino Biological). For typical preparation, 2 L of Hi5 cells at $1.5 \times 10^6$ cells/ml were infected at a multiplicity of infection (MOI) of 3. After 72 h, the cells were collected by centrifugation at 4,000$g$ for 1 h, lysed in 50 ml buffer containing 50 mM Tris-HCl (pH 8.0), 150 mM NaCl, and 1 mM DTT, lysed by ultrasonication, after which the soluble proteins were separated from cell debris by centrifugation at 27,000$g$ at 4˚C for 30 min. The supernatant was loaded onto Ni-NTA (Nickel-Nitrilotriacetic acid) resin (Invitrogen), and PA200 was affinity-purified by using an N-terminal 6 His-tag. PA200 was purified further by gel filtration chromatography using a Superose 6 HR 10/30 column (GE Healthcare) pre-equilibrated with buffer containing 20 mM HEPES (pH 7.5) and 150 mM NaCl, 1 mM DTT. Finally, the purified PA200 was concentrated with a centrifugal filter (Amicon Ultra) to approximately 2 mg/ml, aliquoted and flash-frozen in liquid nitrogen before storing at −80˚C.

### Cryo-EM sample preparation and data acquisition

Human 20S proteasome was purchased from Boston Biochem with a concentration of 1.4 mg/ml in 50 mM HEPES (pH 7.6), 100 mM NaCl, 1 mM DTT. For PA200-20S-CP complex, a 3 μl volume of 20S-CP (approximately 1.4 mg/ml) was incubated with a 6 μl volume of PA200 (approximately 0.44 mg/ml) in ice for 1 h. A 4 μl volume of the sample was applied to a glow-discharged Quantifoil copper grid and vitrified by plunge freezing in liquid ethane using a Vitrobot Mark with blotting time of 3 s identically. Data collection was performed on a Titan Krios microscope operated at 300 kV and equipped with a field emission gun, a Gatan GIF Quantum energy filter, and a Gatan K2 Summit direct electron camera in superresolution mode, at CBI-IBP. The calibrated magnification was 130,000 in EFTEM mode, corresponding to a pixel size of 1.04 Å. The automated software SerialEM was used to collect 1,300 movies at a defocus range of between 1.8 and 2.3 μm. Each exposure (10-s exposure time) comprised 32 subframes, amounting to a total dose of 60 electrons $Å^{-2} s^{-1}$. PA200 cryo sample preparation and data acquisition were performed as above, with 526 snapshots collected.

### Image processing

Micrograph movie stacks were corrected for beam-induced motion using MotionCor2 [61]. The contrast transfer function parameters for each dose-weighting image were determined with Gctf [62]. Particles were initially autopicked with Gautomatch (https://www.mrc-lmb.cam.ac.uk/kzhang/) without template and extracted with a 200 × 200-pixel box for PA200 dataset and a 380 × 380-pixel box for PA200-20S-CP dataset. Reference-free 2D-class average was performed using RELION [63], and the well-resolved 2D averages were subjected to another iteration particle autopicking as a template with Gautomatch. After iterative 2D-class average in RELION, only particles with best-resolved 2D averages were selected for initial model generation and 3D classification using RELION. The classes with identical details that

show good features of PA200 were merged for further autorefinement with a sphere mask and postprocessed with 3-pixel extension and 3-pixel falloff around the entire molecule to produce the final density map with an overall resolution of 3.75 Å for PA200 dataset and 2.92 Å for PA200-20S-CP dataset [64]. To further improve map quality and local features of the PA200 region in PA200-20S-CP and reconstruct uncapped 20S map, we applied the block-based reconstruction method for further classification [65]. To distinguish the uncapped and single-capped complex from the top view (globular particles), we combined all top-view particles with single-capped PA200-20S side-view particles to reconstruct high-resolution structure; then the subparticles, including PA200 and few densities of 20S-CP interacting with PA200, were reextracted from original images by calculating a vector from the center of the whole complex to the center of PA200, using orientation from whole particles' reconstruction. The subparticles were then subjected to another 3D classification without alignment, and only particles with good features of PA200 were selected for the final reconstruction, CTF refinement, and postprocessing [66]. This process yielded a PA200-20S-CP map with overall resolution of 2.72 Å. The subparticles without density of PA200 were reextracted from original images by calculating a vector from the center of PA200 to the center of whole complex and combined with particles that were classified as 20S and bad particles in previews of 2D class average procedure and subjected to 3D classification and refinement. This process produced the final 20S map with an overall resolution of 3.3 Å. To validate the model, the final dataset, about 103,000 and 87,000 particles of PA200 and PA200-20S-CP complex, respectively, were further subjected to cisTEM for initial model generation, 3D autorefinement, and sharpening. The maps generated from RELION and cisTEM independently were consistent [67], indicating no model bias. Chimera and PyMOL (The PyMOL Molecular Graphics System Version 1.8) were used for graphical visualization [68].

## Model building and refinement

Ab initio modeling of PA200 was performed in Coot [69], using structure predictions calculated by Phyre2 [70] and the partial structure modeled by EMBuilder [71]. Model of the native human 20S proteasome was downloaded from PDB (ID: 5LE5) [26]. Map refinement was carried out using Phenix.real_space_refine, with secondary structure and Ramachandran restraints [72].

## Proteasome activity assay

The 20S Proteasome Activity Kit was used to analyze proteasome chymotrypsin-like (β5) activity. The chymotrypsin-like activity of 20S CP or PA200-20S for the fluorogenic peptide substrates (Suc-LLVY-AMC, 1 mM) was assessed by monitoring the kinetics of fluorescence increase of generated free AMC in a plate reader (Spectramax i3x; excitation: 360 nm; emission: 460 nm). The 20S CP and PA200-20S were tested at a final concentration of 2.5 nM. The assay mixture was preincubated for 15 min at 25°C, and an increase in fluorescence was recorded at an interval of the 30 s for 30 min at 32°C. The top panel of Fig 3G shows Time-RFU and was generated by EXCEL software (S1 Data). The middle and bottom panels of Fig 3G show the proteasome activity assays (S2 and S3 Data).

The data were analyzed with GraphPad Prism 6 to calculate the values of Km, Vmax, and $R^2$. (1) For preparation of the standard curve, AMC standard was diluted by 1× proteasome assay buffer to 8 μM, 4 μM, 2 μM, 1 μM, 0.5 μM, 0.25 μM, 0.125 μM, 0 μM. Fluorescence of AMC was tested by Spectramax i3x (excitation: 360 nm; emission: 460 nm). The experiments were done in triplicate, and the fluorescence data were averaged and processed by EXCEL. The RFU-AMC standard curve was shown as y = 417,295x + 43,332. (2) The proteasome

activity of the 20S and PA200-20S was evaluated at a final concentration of 2.5 nM with the dilution of Suc-LLVY-AMC as substrate (20S: 25 μM, 50 μM, 100 μM, 150 μM, 200 μM, 300 μM; PA200-20S: 25 μM, 50 μM, 100 μM, 200 μM, 300 μM, 400 μM, 500 μM). The experiments were done in triplicate. The RFU was tested and transferred to the concentration of AMC by the standard curve. And then, the slopes of all the Time-AMC concentration curves, which represent the enzyme reaction rate of different concentrations of Suc-LLVY-AMC, were obtained. These data were calculated by GraphPad Prism 6 to obtain the Km, Vmax, and $R^2$.

## HPLC-MS

HPLC-MS analysis was performed on the Thermo HPLC-MS system (Thermo Scientific, Waltham, MA, USA) using the Thermo Hypersil GOLD C18 column (1.9-μm particle size, $2.1 \times 100$ mm) with acetonitrile (A) and water (B) as eluents. The experimental conditions were as following: injection volume: 7 μM; mobile phase: 0–1 min, 100% B; 1–4 min, 100%–5% B; 4–9.5 min, 5% B; 9.5–10 min, 5%–100% B; 10–11 min, 100% B; flow rate: 0.3 ml·min$^{-1}$. The HPLC eluate was administered to the MS system with a spray voltage of 1.0 kV. The MS peaks were recorded and compared with that of the Insp6 standard (phytic acid sodium salt hydrate, $C_6H_{18}O_{24}P_6 \cdot xNa^+ \cdot yH_2O$, MERCK, P8810).

## Supporting information

**S1 Fig. Data processing flowchart of PA200 and PA200-20S.** We applied routine processing including classification and refinement to obtain particles' orientation, then reextracted subparticles consisting of whole PA200 and low density of 20S-CP interacting with PA200 by calculating a vector in pixels from the whole complex to the center of PA200. The subparticles were subjected to another 3D classification without alignment, and only particles with good features of PA200 were selected for the final reconstruction, CTF refinement, and postprocessing. CP, core particle; CTF, contrast transfer function; PA200, proteasome activator 200. (TIF)

**S2 Fig. PA200 sequence and its features.** HEAT repeat helices are labeled 1A for helix A of HEAT repeat 1, etc. The BRDL domain of PA200 is colored magenta. The C-terminal (YYA) and the loop (Thr562 to Lys574) inserted into the 20S are colored red and green, respectively. The residues involved in the interaction between PA200 and 5,6[PP]-InsP$_4$ and InsP$_6$ are colored cyan and yellow. The key residues of BRDL are indicated by ☆. 5,6[PP]2-InsP$_4$, (5,6)-bis-diphosphoinositol tetrakisphosphate; BRDL, bromodomain-like; InsP$_6$, inositol hexakisphosphate; PA200, proteasome activator 200. (TIF)

**S3 Fig. The density map of the two α-rings of PA200-free 20S.** (A-B) Close-up views of the two α-rings (A: top end; B: bottom end). The cryo-EM maps are shown as gray mesh and the atomic models as cartoon. Both gates of the two α-rings are closed and indicated with a dotted circle. cryo-EM, cryo–electron microscopy; PA200, proteasome activator 200. (TIF)

**S4 Fig. Close-up representations of cofactors bound by apo PA200 and the PA200 in the complex.** (A-B) Cryo-EM densities bound by the openings of PA200 in the complex are fitted with 5,6[PP]-InsP$_4$ (A, opening 1) and InsP$_6$ (B, opening 2), respectively. (C-D) Cryo-EM densities bound by the openings of apo PA200 are fitted with 5,6[PP]-InsP$_4$ (A, opening 1) and InsP$_6$ (B, opening 2), respectively. 5,6[PP]2-InsP$_4$, (5,6)-bisdiphosphoinositol tetrakisphosphate; cryo-EM, cryo–electron microscopy; InsP$_6$, inositol hexakisphosphate; PA200,

proteasome activator 200.
(TIF)

**S5 Fig. The BRDL domain of Blm10.** (A-B) Same as Fig 6A and 6B, panel A shows the bottom view Blm10 BRDL domain (PDB: 4V7O), whereas panel B shows the side view. (C) Same as Fig 6C, the BRDL domain of Blm10 is shown in red and superpositioned with that of its PA200 counterpart (yellow) in the right black box. The N-terminal of Blm10 is shown as red spheres, whereas the C-terminal is indicated as blue spheres. Blm10, Bleomycin resistance 10; BRDL, bromodomain-like; PA200, proteasome activator 200; PDB, Protein Data Bank. (TIF)

**S6 Fig. Domain organization and overall fold of BRDs.** (A) Schematic drawing of PA200 BRDL domain. Key residues are colored red, whereas a black star indicates the Asn residue that binds acetyl-Lys. (B) Domain organization and the sequence alignment of the typical human BRD families. (C) Ribbon diagrams of eight BRD families and PA200. PDB code: GCN5L2 (3D7C), BRD4(1) (2OSS), EP300 (3I3J), BRD9 (3HME), TIF1α (2YYN), TRIM28 (2RO1), TAF1L(2) (3HMH), PB1(1) (3IU5). BRD, bromodomain; BRDL, BRD-like; PA200, proteasome activator 200; PDB, Protein Data Bank. (TIF)

**S7 Fig. Cryo-EM map of PA200, PA200-20S, and some key regions.** (A) Cryo-EM map (gray mesh) of the recombinant human PA200 with a fitted atomic model. (B) Cryo-EM map (gray mesh) of the recombinant human PA200-20S complex with a fitted atomic model. (C) A rectangular lid covering the region of PA200 previously reported as a big lateral opening ($13 \times 22$ Å) located at the dome-like structure of Blm10. Close-up views of the cryo-EM map (gray mesh) with a fitted atomic model (cartoon representation) of the lid. (D-E) Close-up views of the cryo-EM map (gray mesh) with a fitted atomic model (cartoon representation) for the two openings and inositol phosphate cofactors (D: opening 1, E: opening 2). (F) Close-up views of the cryo-EM map (gray mesh) with a fitted atomic model (cartoon representation) for the BRDL domain. Blm10, Bleomycin resistance 10; BRD, bromodomain; BRDL, BRD-like; cryo-EM, cryo–electron microscopy; PA200, proteasome activator 200. (TIF)

**S1 Table. Data collection and refinement statistics.**
(DOCX)

**S1 Data. Fluorescence data for the top panel of Fig 3G.**
(XLSX)

**S2 Data. 20S fluorescence data for the middle panel of Fig 3G.**
(XLSX)

**S3 Data. The 20S-PA200 fluorescence data for the bottom panel of Fig 3G.** PA200, proteasome activator 200.
(XLSX)

## Acknowledgments

The cryo-EM data were collected at the Center for Biological Imaging (CBI), Institute of Biophysics, CAS.

## Author Contributions

**Funding acquisition:** Hongxin Guan, Ping Zhu, Songying Ouyang.

**Methodology:** Hongxin Guan, Youwang Wang, Ting Yu, Yini Huang, Mianhuan Li, Vanja Perčulija, Daliang Li, Jia Xiao, Dongmei Wang.

**Project administration:** Hongxin Guan, Youwang Wang, Ting Yu, Yini Huang, Daliang Li, Dongmei Wang, Ping Zhu, Songying Ouyang.

**Software:** Youwang Wang, Ting Yu, Mianhuan Li, Abdullah F. U. H. Saeed, Jia Xiao, Ping Zhu.

**Validation:** Songying Ouyang.

**Writing – original draft:** Hongxin Guan, Youwang Wang, Abdullah F. U. H. Saeed.

**Writing – review & editing:** Hongxin Guan, Youwang Wang, Abdullah F. U. H. Saeed, Vanja Perčulija, Ping Zhu, Songying Ouyang.

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
