## [Editor Report · Decision Letter 0]

19 Sep 2019

Dear Dr Ouyang, 

Thank you for submitting your manuscript entitled "Cryo-EM structures of the human PA200 and PA200-20S complex reveal regulation of proteasome gate opening and two novel PA200 apertures" for consideration as a Research Article by PLOS Biology.

Your manuscript has now been evaluated by the PLOS Biology editorial staff as well as by an academic editor with relevant expertise and I am writing to let you know that we would like to send your submission out for external peer review.

Please re-submit your manuscript within two working days, i.e. by Sep 23 2019 11:59PM.

Kind regards,

Ines

--

Ines Alvarez-Garcia, PhD

Senior Editor

PLOS Biology

Carlyle House, Carlyle Road

Cambridge, CB4 3DN

+44 1223–442810

---

## [Decision Letter · Decision Letter 1]

17 Oct 2019

Dear Dr Ouyang,

Thank you very much for submitting your manuscript "Cryo-EM structures of the human PA200 and PA200-20S complex reveal regulation of proteasome gate opening and two novel PA200 apertures" for consideration as a Research Article at PLOS Biology. Your manuscript has been evaluated by the PLOS Biology editors, an Academic Editor with relevant expertise, and by three independent reviewers.

As you will see, the reviewers find your results interesting and significant, however they also raise several points that should be addressed with experiments or further analyses of the data. In addition, they all ask for several clarifications. After discussing the reviews with the Academic Editor, we feel that there are two points that are beyond the scope of this study: 1) effects of mutations to the YYA motif and 2) functional differences of the inositol species (Rev. 1 Points 2 and 3). While we will welcome any data you might have in hand, we won’t make these requests essential for publication.

In light of the reviews (attached below), we will not be able to accept the current version of the manuscript, but we would welcome resubmission of a revised version that takes into account the reviewers' comments. We cannot make any decision about publication until we have seen the revised manuscript and your response to the reviewers' comments. Your revised manuscript is also likely to be sent for further evaluation by the reviewers.

Your revisions should address the specific points made by each reviewer. Please submit a file detailing your responses to the editorial requests and a point-by-point response to all of the reviewers' comments that indicates the changes you have made to the manuscript. In addition to a clean copy of the manuscript, please upload a 'track-changes' version of your manuscript that specifies the edits made. This should be uploaded as a "Related" file type. You should also cite any additional relevant literature that has been published since the original submission and mention any additional citations in your response. 

Before you revise your manuscript, please review the following PLOS policy and formatting requirements checklist PDF: http://journals.plos.org/plosbiology/s/file?id=9411/plos-biology-formatting-checklist.pdf. It is helpful if you format your revision according to our requirements - should your paper subsequently be accepted, this will save time at the acceptance stage.

Please note that as a condition of publication PLOS' data policy (http://journals.plos.org/plosbiology/s/data-availability) requires that you make available all data used to draw the conclusions arrived at in your manuscript. If you have not already done so, you must include any data used in your manuscript either in appropriate repositories, within the body of the manuscript, or as supporting information (N.B. this includes any numerical values that were used to generate graphs, histograms etc.). For an example see here: http://www.plosbiology.org/article/info%3Adoi%2F10.1371%2Fjournal.pbio.1001908#s5.

For manuscripts submitted on or after 1st July 2019, we require the original, uncropped and minimally adjusted images supporting all blot and gel results reported in an article's figures or Supporting Information files. We will require these files before a manuscript can be accepted so please prepare them now, if you have not already uploaded them. Please carefully read our guidelines for how to prepare and upload this data: https://journals.plos.org/plosbiology/s/figures#loc-blot-and-gel-reporting-requirements.

Upon resubmission, the editors will assess your revision and if the editors and Academic Editor feel that the revised manuscript remains appropriate for the journal, we will send the manuscript for re-review. We aim to consult the same Academic Editor and reviewers for revised manuscripts but may consult others if needed.

We expect to receive your revised manuscript within two months. Please email us (plosbiology@plos.org) to discuss this if you have any questions or concerns, or would like to request an extension. At this stage, your manuscript remains formally under active consideration at our journal; please notify us by email if you do not wish to submit a revision and instead wish to pursue publication elsewhere, so that we may end consideration of the manuscript at PLOS Biology.

When you are ready to submit a revised version of your manuscript, please go to https://www.editorialmanager.com/pbiology/ and log in as an Author. Click the link labelled 'Submissions Needing Revision' where you will find your submission record. 

Sincerely,

Ines

--

Ines Alvarez-Garcia, PhD

Senior Editor

PLOS Biology

Carlyle House, Carlyle Road

Cambridge, CB4 3DN

+44 1223–442810

Reviewers’ comments

Rev. 1:

In this manuscript Zhu, Ouyang and colleagues describe the cryo-EM structure of the human PA200 and PA200-20S complexes. They show two openings exist in PA200, which might be suitable to thread proteins destined for degradation into the 20S proteasome. This would be facilitated by the fact that PA200 binding opens the gate of the 20S proteasome and in accordance with this find a higher activity in the degradation of model fluoregenic peptide substrates. Furthermore the authors find inositol phosphates to bind to PA200, which they propose to act as molecular glues that play a role in histone degradation of hyperacetylated histones, such as those found in DNA damage responses. In further support of this hypothesis they find a BRDL in PA200, which is structurally divergent from other family members and akin to yeast Blm10, which may aid in the degradation of acetylated histones.

While the structures determined by the authors appear sound, several aspects of the work needs further experimental underpinning and are not conclusively addressed in the manuscript at present.

1) The authors clam that PA200 is unable to bind both ends of the 20S proteasome even in high molar excesses in Figure1. However, they clearly find double-capped particles in negative stained images, which appear to be lost in the cryo preparation of Figure 1C. In my opinion the authors need to demonstrate if the second PA200 is lost as a consequence of grid preparation. They should show by biochemical means if double capped particles exist and what the propotion of them is in comparison to the EM preparations.

2) With regards to gate opening, they should determine a structure of double-capped PA200-20S to reveal if gate opening is indeed a consequence of PA200 binding or just an inherent assymmetry of the 20S proteasome. Related to this aspect, what effects do mutations of the YYA motif elicit in degradation assays?

3) What is the function of the function of the different inositol species on peptide degradation. Are the inositol phosphates required for the stimulation of proteolytic activity? Does the it have to be a mixture of both inositol phosphates, or is one or the other sufficient? A further question that arises: Could the density for the 5,6(PP)2-InsP4 also be interpreted as InsP6 and two solvent molecules?

4) Regarding the strong implications for histone degradation brought forward by the authors: It appears that experimental evidence should be provided by the authors.

I therefore commend the authors on potentially interesting work, but at the present state feel that the manuscript requires further experimental underpinnings.

Rev. 2:

The paper by Guan et al described the use of cryoEM to solve the structure of PA200 and PA200 bound to the proteasome core particle (20S). Proteasomes are important complexes in the cell and beside the stereotype 26S proteasome, which has 20S associate with 19S regulatory particle and recognizes ubiquitinated substrates, there are other important regulators. This paper describes a refined structure of the human version of one of these regulators, PA200, both by itself and in association with 20S. This work is a refinement of a previous EM structure at much lower resolution and this new high resolution allows for a detailed comparison of this complex with its yeast orthologue Blm10-CP. The latter has been solved at high resolution by X-ray. The authors did a nice job and provided a detailed comparison between the different structures. New insights are the observation of two “channels” in the PA200 dome that sits on the 20S that are blocked/bound with IP4 and IP6. The binding of these compounds is intriguing but the relevance or biology related to it remains unclear. The structural detail provided by this paper is valuable, however, the amount of new insights and biological or mechanistic understanding gained is still limited as there are no experiments to follow up or further strengthen the paper.

It also has to be noted that a very recent recently published paper in Molecular Cell describes the same structure(Oct 3 2019). I don’t think this should be hold against the authors and the publication of this work is a timely confirmation. Also, the focus and description are somewhat different such that this appear has value. That said, I like to point out that both papers provides similar insights, but that the other paper also reveals an allosteric pathway that links the PA200 binding to the active sites which is intriguing. However, I believe the main impact and justification of the other paper in Molecular Cell is the technical advance that that paper described, as it provides promising opportunities for in vitro mutagenesis and other approaches. Namely, the molecular cell paper used a heterologously expressed insect cell system to purify all components and thus for the first time was able to purified the eukaryotic 20S from heterologous expressed source (by expression of the 14 CP subunits + 5 chaperones from one baculo virus construct). This opens the door for mutant analysis of this essential complex. The paper under review here did an excellent effort on purifying PA200 from insect cells (like the Mol Cell), but purchased the 20S and thus lacks that aspect of impact. It is nice to see similar structures from both approaches and I by no means want to diminish the current work, just explain the differences for the editor.

Some specific points:

-From the description and the work it is not fully clear if the identification of IP4 and IP6 is fully confident based on the density maps. Cleary it has not been experimentally addressed or confirmed, so would some caution in the conclusion be valid?

-Line 198. PA200 can interact with three complexes: Does PA200 not associate with immature CP complexes? The yeast orthologue does.

-Line 297 “binds acetylated histones with ..as key residues.” & line 305. Explain in more detail, authors do not show this, they propose this? Or references? Line 313 PA200 blm10 unlikely to bind acetylated histone based on this structure? How is the binding site of the acetylated histone related to the two channels described? Does the structure provide any clues towards the weird proposed ability of PA200-20S to be able to degrade histones?

-The authors describe a PA200 induced increase in Vmax. Do they believe this is from gate opening? Also, the authors ignore the Km change they observe. The change in Km would suggest a change in the substrate binding environment, which would be consistent with the change observed in the recent Molecular cell paper?

The language and description of the paper is overall good, however, some care could be taken to polish it further. E.g. the improper use of significant , see e.g. line 43, line 64.; line 50 replace “especially” with “and particularly high” or something like that. Line 51 complexES. Line 225 has -> have.

Line 209: conformational change not clear from picture to me.

Rev. 3:

In their manuscript Guan et al. describe the analysis of human PA200-20S complexes obtained by in vitro assembly of recombinant PA200 and endogenous 20S. Although their data may be of interest, overall the manuscript could have been better written and there are some major issues that need addressing.

1 - the authors state in the introduction that this manuscript follows the recent publication of another high resolution cryo-EM study of human PA200-20S complexes. Most of the features described in their manuscript have already been presented there. Accordingly, they should revise the presentation of their results in the light of the published data. Starting from the current title, with the reference to the “regulation of proteasome gate opening and two novel PA200 apertures”, the authors refer to features that are presented as new, but which unfortunately have already been described. This continues throughout the manuscript. The authors must clarify what is really the new information obtained from their analysis and acknowledge what is already known.

2 - the authors assembled their PA200-20S complexes by adding recombinant PA200 to an endogenous commercially available 20S. The electron microscope images in Figures 1A and 1B show significant heterogeneity of the samples obtained. Could this be improved by an additional size exclusion step? The authors should also clarify if they expressed PA200 in insect or bacterial cells, as both are referred to in the methods.

3 - Figures 1A and 1B also show that the PA200 and 20S binding efficiency is not very good. Could this be due to biochemical related issues, such as the use of non-fresh 20S? Additionally, since it is well known that particle picking software favors the recognition of globular particles, rather than more elongated ones like PA2002 20S, it is not clear if the different ratios of uncapped, single-capped and double-capped complexes obtained are physiologically significant.

4 - The authors “refined the resolution of PA200-20S complex to 2.72 A via block-based reconstruction by using the threshold value of 0.143 for the gold-standard Fourier shell correlation”. These procedures were developed to deal with Ewald sphere effects in the reconstruction of large particles, such as viruses, and it would be interesting for the reader if the authors would clarify why these procedures were selected. Did the authors use the Relion software for this? Following the Relion workflow, the authors also did post-processing using a mask corresponding to a “3-pixel extension and 3-pixel fall-off around the entire molecule”. Because this is a quite tight and sharp mask could the authors discuss how they avoided overfitting? Why did the maps obtained with Relion needed validation using cisTEM? Could other validation approaches be used?

5 - the authors mentioned they determined the structures of both PA200-20S and PA200 on its own. However, they do not present the structure of PA200 in detail, nor they discuss it including regarding any conformational changes that may occur upon its binding to 20S.

6 - the authors monitored the interaction of PA200 with the 20S by measuring its chymotryptic-like activity. However, others reported that PA200 enhances the tryptic- or caspase-like activities instead. This needs to be discussed.

7- the authors describe a PA200 BRD domain that “can’t be classified into either of the eight typical human BRD families”. This is very much based on the sequence analysis of PA200, but what is the evidence that this novel domain has indeed a BRD function?

8 - the representation of the experimental structures needs significant improvement, to allow the reader to appreciate their quality and the models built. As an example, in Figure 5D a slice of the full map densities should be shown, rather than just showing the densities of interest, so that the quality and strength of the densities for the cofactors could be fully appreciated.

---

## [Decision Letter · Decision Letter 2]

15 Jan 2020

Dear Dr Ouyang,

Thank you for submitting your revised Research Article entitled "Cryo-EM structures of the human PA200 and PA200-20S complex reveal regulation of proteasome gate opening and two PA200 apertures" for publication in PLOS Biology. I have now obtained advice from the original reviewers and have discussed their comments with the Academic Editor. 

Based on the reviews, we will probably accept this manuscript for publication, assuming that you will modify the manuscript to address the remaining points raised by Reviewers 2 and 3. Please also make sure to address the data and other policy-related requests noted at the end of this email.

We expect to receive your revised manuscript within two weeks. Your revisions should address the specific points made by each reviewer. In addition to the remaining revisions and before we will be able to formally accept your manuscript and consider it "in press", we also need to ensure that your article conforms to our guidelines. A member of our team will be in touch shortly with a set of requests. As we can't proceed until these requirements are met, your swift response will help prevent delays to publication.

*Copyediting*

*Published Peer Review History*

*Early Version*

*Submitting Your Revision*

Sincerely,

Ines

--

Ines Alvarez-Garcia, PhD

Senior Editor

PLOS Biology

Carlyle House, Carlyle Road

Cambridge, CB4 3DN

+44 1223–442810

DATA POLICY:

Fig. 3G

Reviewers’ comments

Rev. 1: Ashwin Chari – please note that this reviewer has waived anonymity.

In the revised version of the manuscript, the authors have addressed my concerns and those of the other reviewers. I therefore recommend publication of the revised manuscript and congratulate the authors for a comprehensive and well-conducted study!

Rev. 2:

The authors properly addressed many of the reviewer comments. However, I find the discussion on BRD/BRDL domain utterly confusing and lacking in logic. It is an important issue that should be more clearly described and defined by the authors.

The authors describe that both PA200 and Blm10 have two regions with BRD-like fold according to them (although not recognized by other Blm10 and PA200 structure papers). Neither of the domains however shows sequence homology to the BRD domain. ("In the human PA200, although the 82-residue BRDL domain folds into the same geometry as typical human BRDs, it does not share significant sequence homology with them [20]." ). Nevertheless, they define the first one in PA200 (aa 1650-1731 in PA200) as a BRDL because of the functional study by another group that suggests acetylated Histone binding. The other domain however no functionally studies have been done, so that seems defined arbitrary. Further puzzling is an interesting observation the authors make to a weird conservation (if all is true), because the BRDL of PA200 (aa 1650-1731 in PA200) aligns with aa 1961-2042 region of Blm10 which also has alpha helixes and similar fold there, but this is not a BRD they argue because it the sequence is not conserved. I appreciate the insight that the BRDL domains between PA200 and Blm10 do not align, which is something that was not noted in the 2016 paper I believe and I think is an important (and maybe concerning) point. Blm10 has a BRDL (aa 1650-1732 in Blm10) based on other paper binding study that aligns with region aa 1356-1432 region of PA200 which has folds similar.

In all, I don't think this is presented clearly and logical. Also, PA200 and Blm10 are full of HEAT repeat and some of the alpha helixes in PA200 have been assigned as Heat repeats, how unique is the BRDlike fold in that context within this protein? Without the described acetylated Histone binding, would the authors define/recognize the specific one BRDlike domain? Do we have BRD like domains and remenants of BRDlike domains? Or no BRDlike domains at all?

- Ana is the first name and paper should be referenced to as "Toste Rêgo et al. "

Rev. 3:

Guan et al. have revised their manuscript, and the mass-spectrometry analysis showing the presence inositol phosphates in the PA200 sample is an important new contribution. The authors provided an extended reply to the questions raised by the reviewers. However, the implementation of their replies into the manuscript appears to be not so successful. Regarding the replies to my previous queries:

Q1: I suggested that credit should be given to Toste Rego and da Fonseca, whose manuscript was published in the October's issue of Molecular Cell, for the observations common to both manuscripts. On their reply to the reviewers the authors say that they:

"changed the description of some sentences and cited the Molecular Cell paper when we were analyzing and interpreting our data."

However, their reference [55] (Toste Rego and da Fonseca, 2019) is cited only once (line 426). At the very least this should be added to their new paragraph in the discussion addressing the overlap of the work (lines 440-450). Here, the sentence:

"During our manuscript submission process, another structural study on PA200 and PA200-20S proteasome complex by Ana et al. was published in Molecular Cell."

Should read:

"During our manuscript submission process, another structural study on PA200 and PA200-20S proteasome complex by Toste Rego and da Fonseca was published in Molecular Cell [55]."

Q2 and Q3: In their reply to the reviewers the authors provided adequate answers for these questions, regarding sample heterogeneity and ratios between the different complexes observed in their analysis. However, this information should also be added to the manuscript, even if briefly in the Methods session, so that these points would be clearer to all manuscript readers.

Q4: Again, the content of the reply to the reviewers regarding the image processing should have been added to the manuscript Methods.

Q6: The authors present a lengthy reply to this question, related to the activity of the PA200-20S and the apparent discrepancy with other studies. The fact is that such discrepancies may be simply related with differences in sample preparation, which differ significantly between the different studies. What is surprising is that, taking into account that others reported changes in the tryptic- and caspase-like activities, the authors only tested the chymotryptic-like activity of their PA200-20S. The three activities should have been tested.

Q7: the points regarding the BRD domain appear to be speculated from existing information, and for this reason it may be better placed in the manuscript Discussion rather than its Results.

---

## [Editor Report · Decision Letter 3]

21 Feb 2020

Dear Dr Ouyang,

On behalf of my colleagues and the Academic Editor, Kylie J. Walters, I am pleased to inform you that we will be delighted to publish your Research Article in PLOS Biology. 

Early Version

PRESS 

Kind regards,

Vita Usova

Publication Assistant, 

PLOS Biology

on behalf of

Ines Alvarez-Garcia,

Senior Editor

PLOS Biology